# Topological dynamics of adiabatic cat states

**Jacquelin Luneau[1,2,3,4⋆], Benoît Douçot[5] and David Carpentier[1]**

1 ENSL, CNRS, Laboratoire de Physique, F-69342 Lyon, France
2 Technical University of Munich, TUM School of Natural Sciences,
Physics Department, 85748 Garching, Germany
3 Walther-Meißner-Institut, Bayerische Akademie der Wissenschaften,
85748 Garching, Germany
4 Munich Center for Quantum Science and Technology (MCQST),
80799 Munich, Germany
5 Laboratoire de Physique Théorique et Hautes Energies, Sorbonne Université
and CNRS UMR 7589, 4 place Jussieu, 75252 Paris Cedex 05, France

⋆ jacquelin.luneau@wmi.badw.de

## Abstract

We consider a quantum topological frequency converter, realized by coupling a qubit to two slow harmonic modes. The dynamics of such a system is the quantum analog of topological pumping. Our quantum mechanical description shows that an initial state generically evolves into a superposition of two adiabatic states. The topological nature of the coupling between the qubit and the modes splits these two components apart in energy: for each component, an energy transfer at a quantized rate occurs between the two quantum modes, in opposite directions for the two components, reminiscent of the topological pumping. We denote such a superposition of two quantum adiabatic states distinguishable through measures of the modes' energy an adiabatic cat state. We show that the topological coupling enhances the entanglement between the qubit and the modes, and we unveil the role of the quantum or Fubini-Study metric in the characterization of this entanglement.


# 1  Introduction

Traditionally, a pump is a device that transfers energy from a source - an engine - to a fluid. The transfer is achieved through a suitably designed mechanical coupling. Topological pumping is a modern extension of such a device. The first historical example of such a topological pump was provided by Thouless who considered a periodically modulated in time one dimensional crystal [1, 2], realized experimentally in various forms [3–14]. A simpler realization was provided recently in the form of a qubit driven at two different frequencies [15], later extended to more complex driven zero dimensional devices [16–22]. These studies discussed pumping through the dynamics of the pump described as a driven quantum system. Indeed, the slow and periodically modulated parameters generates a quantized current either through the quantum system, or, in the last case, in an abstract space of harmonics of the drives. This current thereby effectively describes a transfer between the drives.

In this paper, we reconcile the description of such a topological device with that of traditional pumps by considering on equal footing the qubit and the drives. This amounts to describe these drives as quantum mechanical degrees of freedom instead of classical parameters. We model them as quantum modes, characterized by a pair of conjugated operators accounting for their phase and number of quanta. This extends previous mixed quantum-classical descriptions of the drives [23–25], into the concept of topological coupling between a qubit and two quantum modes. We focus on the regime of adiabatic dynamics in which these quantum modes are slow compared to the qubit, neglecting effects induced by a faster drive, recently discussed in the context of topological Floquet systems [26–32].

The dynamics of a system with slow and fast degrees of freedom has been largely studied in terms of an effective Hamiltonian [33–35]. Here we focus on the adiabatically evolved quantum states. Our quantum mechanical description shows that an initial separable state generically evolves into a superposition of two components. Each component is an adiabatic state of the total system. The topological nature of the coupling between the qubit and the modes splits these two components apart in energy at a quantized rate: for each component, an energy transfer occurs between the two quantum modes, in opposite directions for the two components. This topological dynamics effectively creates an adiabatic cat state: a superposition of two quantum adiabatic states distinguishable through measures of the modes' energy. Note that our definition of cat states does not require an equal superposition of two adiabatic components. We quantify the relative weight of each component in terms of the quantum geometry of adiabatic states. While some specific initial states were shown to evolve into cat states in [23], here we show that cat states are indeed generic. On the technical side, we develop an adiabatic approximation method valid to all orders in the modes' frequencies, which shows that the topological splitting of each cat component at a quantized rate is robust at all orders in the adiabatic parameter. This contrasts with the quantization of pumping occuring only for well prepared initial states [36].

Besides, we show that for each of the component of a cat state, the qubit is entangled with the two modes. The origin of this entanglement lies in the geometrical properties of the coupling, characterized by the quantum or Fubini-Study metric [34, 37], related to various physical observables, see *e.g.* [38–46]. Furthermore, the topological nature of the coupling constrains these geometrical properties: a topological coupling necessarily induces a strong entanglement between a pump and its driving modes.

While the mechanism of topological frequency conversion was introduced in [15] as an anomalous dynamics of a qubit driven at two frequencies, it was later discussed within a unified framework of topological pumping as a coupling between a qubit and two slow *classical modes* in [24]. This classical-quantum approach was extended in [23, 25] where one of the two modes was described by a quantum harmonic oscillator, the other one remaining a Floquet drive. In contrast, our approach relies on a complete quantum description of both modes, crucial to describe the decomposition of an initial state into a cat state. On a similar register, in the case of a single quantum mode, the impact of the quantum nature of a mode on a spin-1/2 Berry's phase was discussed in [47]. Interestingly, similar qubit-modes systems were recently proposed [48] and experimentally realized in quantum optical devices [49] to simulate topological lattice models. In this context, the Hamiltonians of a qubit coupled to cavities was expressed in terms of Fock-state lattices, and shown, with two cavities, to realize a chiral topological phase [50, 51], and, with three cavities, the quantum or valley Hall effect [48, 52]. Indeed, the focus of these realizations was on synthetic topological models and their associated zero-energy states. Our approach bridges the gap between the study of topological pumping of driven systems and these studies of quantum optical devices.

Our paper is organized as follows. First, Sec. 2 is a self-contained and synthetic section in which we introduce the model of a qubit topologically coupled to two quantum modes and describe the essential results summarized on Fig. 3. We describe the modes firstly as harmonic oscillators (Sec. 2.1.1) and then introduce the approximation of quantum rotors (Sec. 2.1.2). We then discuss qualitatively the typical dynamics of adiabatic cat states (Sec. 2.2). The more technically interested reader can then turn into Sec. 3 and Sec. 4. In Sec. 3, we develop an adiabatic theory of the rotor model to characterize the two components of the cat states as adiabatic states and identify their topological dynamics which splits them apart in energy. We show that due to the quantum nature of the mode, any realistic initial state decomposes into such a sum of two adiabatic states which split into a cat state. We characterize the weight of each cat component for a separable initial state, and relate it to the quantum geometry of the qubit adiabatic states. In Sec. 4.1, we relate the entanglement between the qubit and the modes to the quantum geometry of the adiabatic states. We unveil the role of the quantum metric in this entanglement and show that a topological coupling is associated to a high entanglement. Finally, we discuss the evolution of the number of quanta of each mode around the quantized drift in relation with Bloch oscillations and Bloch breathing in Sec. 4.2. We show that the quantum fluctuations of the modes reduce the temporal oscillations of the number of quanta around the drift, stabilizing the rate of splitting of the cat components.

## 2 A qubit topologically coupled to two quantum modes

### 2.1 Topological coupling

#### 2.1.1 Two slow harmonic oscillators coupled to a qubit

We consider two quantum harmonic oscillators coupled to a two-level system, a qubit. In the following, we denote by "mode" each harmonic oscillator, by analogy with electrodynamics. The annihilation and creation operators $\hat{a}_i$, $\hat{a}_i^\dagger$, $i = 1, 2$, of the two modes satisfy $[\hat{a}_i, \hat{a}_j^\dagger] = \delta_{ij}\mathbf{1}$. The Hamiltonian of the qubit coupled to the two modes reads

$$\hat{H}_{\text{tot}} = \hbar\omega_1\hat{a}_1^\dagger\hat{a}_1 + \hbar\omega_2\hat{a}_2^\dagger\hat{a}_2 + \hat{H}, \tag{1}$$

where $\hat{H}$ contains the bare Hamiltonian of the qubit and the coupling to the modes. Generically, we decompose this Hamiltonian on the Pauli matrices of the qubit, considering that each of them can couple to the quadratures of the modes,

$$\hat{H} = \sum_{\alpha=x,y,z} h_\alpha(\hat{a}_1, \hat{a}_1^\dagger, \hat{a}_2, \hat{a}_2^\dagger)\sigma_\alpha. \tag{2}$$

This is represented schematically on Fig. 1(a).

Our definition of topological coupling between slow and fast quantum systems is an extension of the topological characterization of the dynamics of the qubit driven by two classical modes [24]. Here the natural slow variables are the phase and number of quanta of the two harmonic oscillators, and the fast degree of freedom is the qubit. To obtain a classical-quantum description of our model, we replace the operators $\hat{a}_i$ and $\hat{a}_i^\dagger$ respectively by the classical variables $\sqrt{n_i}e^{i\phi_i}$ and $\sqrt{n_i}e^{-i\phi_i}$, where the phase $\phi_i$ and number of quanta $n_i$ satisfy the classical Poisson bracket relation $\{\hbar n_i, \phi_j\} = \delta_{ij}$. The quantum version of this classical-quantum description, on which we will focus in the following, is obtained using a Wigner-Weyl phase space representation [53]. In doing so, we obtain the following Hamiltonian of the qubit parametrized now by the phase space variables of the modes

$$H(\phi_1, \phi_2, n_1, n_2) = \mathbf{h}(\phi_1, \phi_2, n_1, n_2) \cdot \sigma, \tag{3}$$

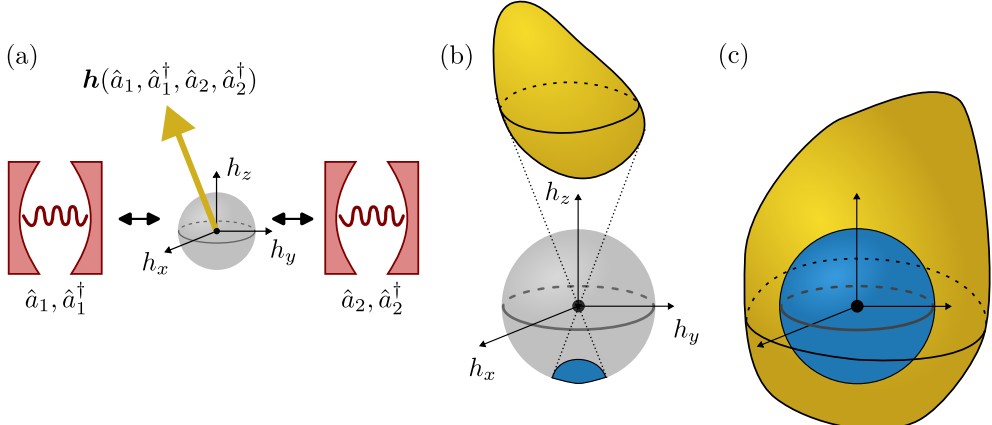

Figure 1: Topological coupling. (a) Two quantum harmonic oscillators, represented by cavities, coupled to a two-level system, represented by a Bloch sphere. The vector $\mathbf{h}(\hat{a}_1, \hat{a}_1^\dagger, \hat{a}_2, \hat{a}_2^\dagger)$ represents the coupling between the Pauli matrices of the qubit and the creation and annihilation operators of the modes, such that the total Hamiltonian reads $\hat{H}_{\mathrm{tot}} = \hbar\omega_1 \hat{a}_1^\dagger \hat{a}_1 + \hbar\omega_2 \hat{a}_2^\dagger \hat{a}_2 + \sum_{\alpha=x,y,z} h_\alpha(\hat{a}_1, \hat{a}_1^\dagger, \hat{a}_2, \hat{a}_2^\dagger)\sigma_\alpha$. (b) We describe each mode by a phase $\phi_i$ and a number of quanta $n_i$, $\hat{a}_i \to \sqrt{n_i}e^{i\phi_i}$, an approximation valid for large enough filling $n_i$. The yellow surface is spanned by the vector $\mathbf{h}(\phi_1, \phi_2, n_1, n_2)$ as the phases $\phi_1, \phi_2$ are varied in $[0, 2\pi]$, at fixed numbers of quanta $(n_1, n_2)$. For standard couplings this surface does not enclose the origin, and the qubit ground states is localized on a restricted region of the Bloch sphere represented as a blue surface. (c) Topological coupling: the surface encloses the origin. Any point of the Bloch sphere corresponds to a ground state of the two-level system for a specified value of the phases of the modes.

with $\mathbf{h}(\phi_1, \phi_2, n_1, n_2)$ a vector of $\mathbb{R}^3$ parametrized by the phases and numbers of quanta of the modes. At fixed value of the number of quanta $n_1, n_2$, we recover a quantum system coupled to two periodic phases $\phi_1, \phi_2$ for which a regime of topological coupling is characterized by a non-vanishing Chern number as in [24]. We provide below a simple condition for this Chern number to be non-zero.

We consider fixed values of $n_1, n_2$ and vary the periodic phases $\phi_1, \phi_2$ in their compact configuration space $[0, 2\pi] \times [0, 2\pi]$. When doing so, the ensemble of vectors $\mathbf{h}(\phi_1, \phi_2, n_1, n_2)$ span a closed surface in $\mathbb{R}^3$. Such a surface is represented schematically on Fig. 1(b). The size of the surface is set by the amplitude of the couplings between the qubit and the phases. To keep a slow-fast separation, the qubit characteristic frequency – given by its gap $|\mathbf{h}|$ – must be large compared to the modes' frequencies $\omega_i$. As such, the closed surface must not touch the origin. The two topologically distinct classes of couplings correspond to whether or not the surface encloses the origin.

In the usual case where the qubit bare transition frequency $\omega_q$ is very large compared to these couplings, the surface is high along the $z$ direction and does not enclose the origin. This is the situation of topologically trivial couplings, represented on Fig. 1(b). The ensemble of ground states of the qubit at fixed $(n_1, n_2)$ is represented on the Bloch sphere by the projection of the surface, represented in blue localized near the south pole. The ensemble of excited states is localized near the north pole, such that we can determine whether the qubit is in its ground or excited state without any knowledge on the state of the modes. This is the common situation of a weak coupling.

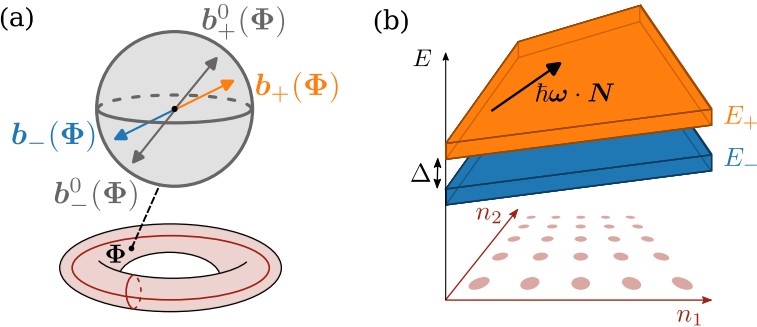

Figure 2: *Phase and number representations of a quantum qubit-2 modes model.*
(a) Phase representation, convenient to represent the dynamics of the qubit. At
each value of the phases $\mathbf{\Phi}$ are associated qubit eigenstates represented by a vector
$\mathbf{b}_\pm^0(\mathbf{\Phi}) = \pm\mathbf{h}(\mathbf{\Phi})/|\mathbf{h}(\mathbf{\Phi})|$ on the Bloch sphere (in grey). The adiabatic states, represented by a vector $\mathbf{b}_\pm(\mathbf{\Phi})$, are a perturbative deformation of the eigenstates. (b)
Number representation, convenient to represent the dynamics of the modes. In this
viewpoint the model can be interpreted as an unusual model of spin-half particle on
a discrete lattice where $\mathbf{N} = (n_1, n_2) \in \mathbb{Z}^2$ represents its position on the lattice and
$\mathbf{\Phi} \in [0, 2\pi]^2$ the Bloch momenta. This particle is submitted to an electric field $\hbar\omega \cdot \hat{\mathbf{N}}$
and a strong spin-orbit coupling $\mathbf{h}(\hat{\mathbf{\Phi}}) \cdot \sigma$. As a consequence, the adiabatic states
are associated to energy bands $E_\pm$ tilted in the direction $\omega$ of the electric field and
separated by the gap $\Delta$ due to the spin-orbit coupling.

In contrast, the topological coupling corresponds to the case where the surface encloses the
origin, represented on Fig. 1(c). Then any point on the Bloch sphere can correspond either to a
ground or excited state, depending on the state of the modes. There is no relevant notion of
qubit ground or excited state independently of the state of the modes. The topological coupling
is a regime of strong coupling, in the sense explained above: the couplings of the qubit to the
phases have to be of same order of magnitude as the bare qubit frequency. This picture also
shows that a topological coupling requires to couple all three Pauli matrices of the qubit to the
quadratures of the modes, *i.e.* the slow modes have to couple to non-commuting observables
of the fast quantum system. This notion of topological coupling is closely related to conical
intersections studied in molecular physics [54–56].

### 2.1.2 Model of quantum rotors

In the following, we describe the two quantum harmonic oscillators coupled to the qubit by
a simpler model of quantum rotors. This amounts to neglecting the dependence in $n_1, n_2$ of
the coupling between the qubit and the modes in the Hamiltonian (3). More precisely, we
consider an initial state of the total system with an average number of quanta $n_i^0$ of each
mode $i = 1, 2$. Ignoring the dependence of the qubit Hamiltonian on the number of quanta
amounts to consider an initial state of spread $\Delta n_i$ such that $\Delta n_i \ll n_i^0$. For the same reason,
this model is valid on short times of the dynamics, as long as the variation of the number
of quanta is small compared to its initial value $n_i^0$. As such, each slow mode is described by
a phase operator $\hat{\phi}_i$ of continuum spectrum $[0, 2\pi]$, conjugated to a number operator $\hat{n}_i$ of
discrete spectrum $\mathbb{Z}$, such that $[\hat{n}_i, \hat{\phi}_j] = i\delta_{ij}$ [57]. Noting $\omega = (\omega_1, \omega_2)$ the frequencies of the
modes, and $\hat{\mathbf{N}} = (\hat{n}_1, \hat{n}_2)$ their respective number operators, the dynamics of the full quantum

system is governed by the Hamiltonian

$$\hat{H}_{\text{tot}} = \hbar\omega \cdot \hat{\mathbf{N}} \otimes \mathbb{I} + H(\hat{\boldsymbol{\Phi}}), \; H(\hat{\boldsymbol{\Phi}}) = \sum_{\alpha=1}^{3} h_\alpha(\hat{\boldsymbol{\Phi}}) \otimes \sigma_\alpha, \quad (4)$$

where the vector $\mathbf{h}(\boldsymbol{\Phi})$ is given by (3) with $\mathbf{N} = \mathbf{N}^0$. The specificity of the rotor model is the linearity of the total Hamiltonian in $\hat{\mathbf{N}}$. We show below that it enables to describe the full quantum dynamics using features of a classically driven qubit, where time-dependent parameters $\phi_i(t) = \omega_i t$ of the qubit Hamiltonian $\mathbf{h}(\phi_1(t), \phi_2(t)) \cdot \sigma$ are here considered as true quantum degrees of freedom.

The modes' intrinsic energies depend on the number of quanta $\mathbf{N}$ while the qubit's energy depend on their phases: hence we will use two dual representations of the dynamics of the system through this paper. When focusing on the qubit's evolution, the phase representation is natural, and represented in Fig. 2(a). At each value of the phases $\boldsymbol{\Phi}$ are associated two qubit's eigenstates $\left|\psi_\pm^0(\boldsymbol{\Phi})\right\rangle$. The modes' dynamics translate into an evolution with time of the phase, and thus an evolution of the associated qubit's states $|\psi_\pm(\boldsymbol{\Phi})\rangle$ which slightly differ from the eigenstates and will be discussed in section 3.1. If instead we focus on the quantum modes, their dynamics is conveniently represented in number representation, Fig. 2(b). The evolution in number representation of the modes' states is solely due to the coupling of the modes to the qubit. Furthermore, in this viewpoint, we can interpret the model as that of a particle on a 2D lattice of sites $\mathbf{N} = (n_1, n_2)$, with $\boldsymbol{\Phi} = (\phi_1, \phi_2)$ being the associated Bloch momentum in the first Brillouin zone. The Hamiltonian (4) describes the motion of this particle, submitted to both a spin-orbit coupling $H(\hat{\boldsymbol{\Phi}})$ and an electric field $\hbar\omega$. We will use this analogy to relate the geometrical and topological properties of the above quantum model to those of gapped phases of particles on a lattice. On a side note, it is useful to notice that in the present case, there is no embedding of the lattice in $\mathbb{R}^2$ as opposed to the Bloch theory of crystals. The position operator identifies with coordinate operator on the lattice. As a consequence, there is no ambiguity in a choice of Bloch convention and definition of the Berry curvature [58,59].

Throughout this paper, the numerical results are obtained by considering an example of such a topological coupling provided by the quantum version of the Bloch Hamiltonian of the half Bernevig-Hughes-Zhang (BHZ) model [60]:

$$h_x(\hat{\boldsymbol{\Phi}}) = \frac{\Delta}{2} \sin(\hat{\phi}_1), \quad (5a)$$

$$h_y(\hat{\boldsymbol{\Phi}}) = -\frac{\Delta}{2} \sin(\hat{\phi}_2), \quad (5b)$$

$$h_z(\hat{\boldsymbol{\Phi}}) = \frac{\Delta}{2}\left(1 - \cos(\hat{\phi}_1) - \cos(\hat{\phi}_2)\right), \quad (5c)$$

where the parameter $\Delta > 0$ is the gap of the qubit. The Chern numbers of the ground and excited bands of this model are $\mathcal{C}_\pm = \mp 1$. We consider modes with frequency of the same order of magnitude, small compared to this gap to ensure the slow-fast separation, $\hbar\omega_1 = 0.075\Delta$ and $\omega_2/\omega_1 = (1+\sqrt{5})/2 \simeq 1.618$. In the following, time is arbitrarily expressed in unit given by the period of the first mode $T_1 = 2\pi/\omega_1$. See appendix C for details on the numerical method.

We focus on the unitary dynamics of a qubit topologically coupled to two quantum modes, valid on timescales smaller that their decoherence and decay times. The effects of dissipation on the Berry curvature of a classically driven two-level system along the lines of [61] are an interesting perspective.

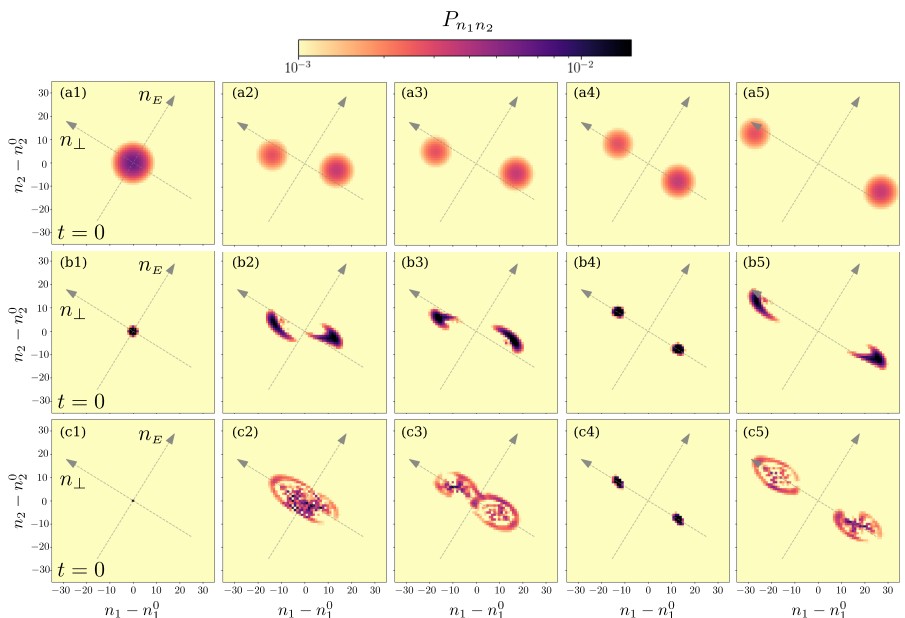

Figure 3: Typical dynamics of adiabatic cat states. Distribution of number of quanta of the two modes $P_{n_1 n_2} = \langle n_1, n_2 | \hat{\rho}_{12} | n_1, n_2 \rangle$ at different times for three initial states. The modes are prepared in a Gaussian state with an average value of phase $\phi_1^0 = \phi_2^0 = 0$ and an equal width in number of quanta $\Delta n_1 = \Delta n_2 = \Delta n$, corresponding to a width $\Delta \phi = 1/(2\Delta n)$ in phase. The qubit is prepared in $(|\uparrow\rangle + |\downarrow\rangle)/\sqrt{2}$. The evolution of states with different initial width $\Delta n$ is represented: line (a), $\Delta n = 5$, $\Delta \phi \simeq 0.03\pi$; line (b), $\Delta n = 0.7$, $\Delta \phi \simeq 0.23\pi$, and line (c), Quasi-Fock state $\Delta n = 1/(2\pi)$, delocalized in phase $\Delta \phi = \pi$. The columns (2) to (5) represent the time evolved state at respectively $t = 8/3, 16/3, 8$ and $t = 32/3$ in units of the period of the first mode $T_1 = 2\pi/\omega_1$. The dynamics splits the initial state in a cat state in the sense of a superposition of two states with distinguishable energy content.

**Relation with Thouless topological pump.** As shown in [24], the topological frequency conversion and the Thouless pumping are two equivalent dynamical phenomena induced by a topological coupling between slow and fast degrees of freedom. In particular, the model (4) can be written as a Rice-Mele model [62], on which most of the experimental realizations of Thouless pumps are based [2]. Such a Thouless pump model is recovered by viewing $\phi_2$ as the time-periodic driving phase of the pump, whereas $n_1, \phi_1$ play the respective role of the unit-cell position and Bloch momentum of the one dimensional lattice. Equivalently, $e^{i\hat{\phi}_1}$ is the translation operator by one unit-cell. Here the qubit encodes the two intra-cell degrees of freedom of the Rice-Mele model. In the Hamiltonian (5), after a $\pi/2$ rotation of the Bloch sphere around the $x$-axis, $-\frac{\Delta}{2}\sin(\phi_2)$ becomes the time-periodic staggered potential while $\frac{\Delta}{2}\cos(\phi_2)$ corresponds to the time-periodic dimerization of the hopping amplitudes of the Rice-Mele model. Hence, our results also describe the effect of the quantum nature of a drive on a Thouless pump.

## 2.2 Topological dynamics of adiabatic cat states

In this section, we first illustrate the topological dynamics of the system starting from a typical state. This dynamics is then analyzed quantitatively in the remaining of this paper. We focus

on separable initial state, easier to prepare experimentally:

$$|\Psi(t=0)\rangle = |\chi_1\rangle \otimes |\chi_2\rangle \otimes |\psi_q\rangle . \tag{6}$$

Each quantum mode is prepared in a Gaussian state $|\chi_i\rangle$,[1] characterized by an average number of quanta $n_i^0$ and a phase $\phi_i^0 = 0$, with widths $\Delta n_i, \Delta \phi_i$ satisfying $\Delta \phi_i \Delta n_i = \frac{1}{2}$. The qubit is prepared in a superposition $|\psi_q\rangle = (|\uparrow_z\rangle + |\downarrow_z\rangle)/\sqrt{2}$.

In Fig. 3, we represent the dynamics of this state $|\Psi(t)\rangle$ by displaying the associated number distribution $P_{n_1 n_2}(t) = \langle n_1, n_2 | \hat{\rho}_{12}(t) | n_1, n_2 \rangle$, with $\hat{\rho}_{12}(t)$ the reduced density matrix of the modes. Three initial states are shown in Fig. 3 (a1), (b1) and (c1) with respective number width $\Delta n = \Delta n_1 = \Delta n_2 = 5$, $\Delta n = 0.7$ and $\Delta n = 1/(2\pi)$ (quasi-Fock state delocalized in phase $\Delta \phi = \pi$). The corresponding time evolved states are represented on columns 1 to 5 of Fig. 3 for times $t = 0, 8/3, 16/3, 8$ and $t = 32/3$. We observe a splitting of the initial state into a superposition of two states

$$|\Psi(t)\rangle = |\Psi_-(t)\rangle + |\Psi_+(t)\rangle . \tag{7}$$

The number distribution of $|\Psi_-(t)\rangle$ and $|\Psi_+(t)\rangle$ drift in opposite directions, corresponding to energy transfers between modes 1 and 2 in opposite directions. This drift is a manifestation of the topological pumping previously discussed within a classical-quantum description [15, 24]. This pumping is conveniently represented by introducing rotated number coordinates

$$n_E = \frac{1}{|\omega|}(\omega_1 n_1 + \omega_2 n_2), \quad n_\perp = \frac{1}{|\omega|}(-\omega_2 n_1 + \omega_1 n_2), \tag{8}$$

with $|\omega| = \sqrt{\omega_1^2 + \omega_2^2}$. $n_E |\omega|$ corresponds to the total energy of the modes and is constant up to the instantaneous energy exchange with the qubit. $n_\perp$ is the coordinate in the direction perpendicular to $\omega$. A transfer of energy between mode 1 and mode 2 occurs as a drift in the $n_\perp$ direction. We also observe that for small initial $\Delta n$, corresponding to lines (b) and (c) of Fig. 3, each component $|\Psi_\pm(t)\rangle$ undergoes a complex breathing dynamics around the drift. This oscillatory behavior is reminiscent of Bloch oscillations of a particle submitted to an electric field, superposed with a topological drift originating from the anomalous transverse velocity. After an initial time of separation, the number distributions for $|\Psi_-(t)\rangle$ and $|\Psi_+(t)\rangle$ no longer overlap (Fig. 3 columns 4 and 5). The system is then in a cat state: a superposition of two states with well distinguishable energy content.

We will now study quantitatively these cat states and their dynamics. In the following section, we develop an adiabatic theory of the rotor model valid at all orders in the modes' frequencies to characterize the two components of the cat states as adiabatic states and identify their topological splitting.

## 3 Adiabatic decomposition

When the driving frequencies remain small compared to the qubit's gap, $\hbar \omega_i \ll \Delta$, we naturally describe the effective dynamics of the coupled qubit and drives in terms of fast and slow quantum degrees of freedom. This is traditionally the realm of the Born-Oppenheimer approximation. Historically both degrees of freedom were those of massive particles, the slow modes being associated with the heavy nucleus of a molecule and the fast ones with the light electrons [33, 63, 64]. In this context, the Born-Oppenheimer approximation assumes that the time evolved

---

[1] For quantum harmonic oscillators, a coherent state $|\alpha\rangle$, $\alpha = \sqrt{n_i^0} e^{i\phi_i^0}$, with an average number of quanta $n_i^0 \gg 1$ reduces to a Gaussian state with $\Delta n_i = \sqrt{n_i^0}$.

state stays close to the instantaneous eigenstates of the fast degrees of freedom and describes the resulting effective dynamics of the slow degree of freedom. For a general slow-fast decomposition of a quantum system, the equations of motion of this effective dynamics involves the Berry curvature of eigenstates of the slow subsystem [35, 65]. These equations of motion govern the dynamics of specific initial states, the adiabatic states, which are defined by an adiabatic projector [65, 66]. The nature of these adiabatic states is often overlooked in the literature. Here we show that they are not naturally prepared experimentally, but any initial state decomposes into a pair of such states. The topological dynamics separates the two components in energy at a quantized rate, leading to a cat state.

## 3.1 Adiabatic projector

The distinctive characteristic of the present quantum rotor - qubit model from the usual Born-Oppenheimer setting is the linearity of the Hamiltonian (4) in the variable $\hat{\mathbf{N}}$. This allows to express in a simple form the corrections to the Born-Oppenheimer approximation for the adiabatic states. The linearity in $\hat{\mathbf{N}}$ enables to write the adiabatic states of the total system in terms of states of the qubit parametrized by the conjugated phases $\boldsymbol{\Phi}$. Let us explain the procedure to construct such states, while referring to appendix A for technical details.

**Adiabatic states of the qubit.** We denote by $\left|\psi_\nu^0(\boldsymbol{\Phi})\right\rangle$, $\nu = \pm$, the normalized eigenstates of the two-level driven Hamiltonian $H(\boldsymbol{\Phi})\left|\psi_\nu^0(\boldsymbol{\Phi})\right\rangle = \nu|\mathbf{h}(\boldsymbol{\Phi})|\left|\psi_\nu^0(\boldsymbol{\Phi})\right\rangle$ for each $\boldsymbol{\Phi}$ in $[0, 2\pi]^2$. For small frequencies $\omega_i$, we can reasonably expect that the qubit-modes system prepared in an eigenstate $|\boldsymbol{\Phi}\rangle \otimes \left|\psi_\nu^0(\boldsymbol{\Phi})\right\rangle$ will remain in a translated eigenstate. However this simple picture is only qualitatively valid: eigenstates get hybridized by adiabatic dynamics, even at arbitrarily small driving frequencies. As a consequence, we first need to identify the family of *adiabatic states* $|\psi_\nu(\boldsymbol{\Phi})\rangle$, stable under the dynamics. In other words, the dynamics is represented as a transport within this family, from $|\boldsymbol{\Phi}\rangle \otimes |\psi_\nu(\boldsymbol{\Phi})\rangle$ to $\left|\boldsymbol{\Phi}'\right\rangle \otimes \left|\psi_\nu(\boldsymbol{\Phi}')\right\rangle$, indexed by translations of the phases $\boldsymbol{\Phi} \rightarrow \boldsymbol{\Phi}' = \boldsymbol{\Phi} - \omega t$. Introducing the dimensionless perturbative parameter $\epsilon$ by $\omega = \epsilon \boldsymbol{\Omega}$, the qubit adiabatic states are a dressing of the eigenstates $\left|\psi_\nu^0(\boldsymbol{\Phi})\right\rangle$ in an asymptotic series of $\epsilon$, see appendix A for details.

**Adiabatic grading of the Hilbert space.** The above adiabatic decomposition of the qubit states allows for a natural decomposition of all states of the whole system. The two corresponding adiabatic projectors are given by

$$\hat{P}_\nu = \int \mathrm{d}\boldsymbol{\Phi} \ |\boldsymbol{\Phi}\rangle \langle\boldsymbol{\Phi}| \otimes |\psi_\nu(\boldsymbol{\Phi})\rangle \langle\psi_\nu(\boldsymbol{\Phi})| \,, \qquad \nu = \pm. \tag{9}$$

They provide a decomposition of any state $|\Psi\rangle$ into two adiabatic states

$$|\Psi\rangle = |\Psi_-\rangle + |\Psi_+\rangle \,, \qquad |\Psi_\nu\rangle = \hat{P}_\nu |\Psi\rangle \,, \ \nu = \pm. \tag{10}$$

Besides, each adiabatic component $|\Psi_\nu\rangle$ can be characterized by a wave amplitude $\chi_\nu(\boldsymbol{\Phi})$ deduced from the wavefunction of the modes in the initial state $\chi(\boldsymbol{\Phi}) = \langle\phi_1|\chi_1\rangle \langle\phi_2|\chi_2\rangle$ according to

$$|\Psi_\nu\rangle = \int \mathrm{d}^2\boldsymbol{\Phi} \ \chi_\nu(\boldsymbol{\Phi})|\boldsymbol{\Phi}\rangle \otimes |\psi_\nu(\boldsymbol{\Phi})\rangle \,, \qquad \chi_\nu(\boldsymbol{\Phi}) = \chi(\boldsymbol{\Phi})\left\langle\psi_\nu(\boldsymbol{\Phi})\middle|\psi_q\right\rangle. \tag{11}$$

The decomposition (10) splits the total Hilbert space $\mathcal{H}_{\mathrm{tot}}$ in two adiabatic subspaces $\mathcal{H}_{\mathrm{tot}} = \mathcal{H}_- \oplus \mathcal{H}_+$, where $\mathcal{H}_-$ and $\mathcal{H}_+$ are respectively the images of the projectors $\hat{P}_-$ and $\hat{P}_+$. Let us stress that this decomposition of the Hilbert space is not a spectral decomposition as the adiabatic projector is not a spectral projector of the total Hamiltonian.

Let us comment on the relation between the adiabatic state (11) and the initial states considered for topological pumps within a classical description of the modes (or at least one of them) [15, 23–25]. In such a description, the slow modes have initially a given phase $\Phi$ and the fast quantum system is prepared in a corresponding eigenstate $\left|\psi_\nu^0(\Phi)\right\rangle$. There are two main differences between this hybrid description and the present adiabatic state (11). First, the quantum nature of the modes leads to a spread in phase given by the wave amplitude $\chi_\nu(\Phi)$. As a result, (11) is not a separable state but is strongly entangled between the modes and the qubit, and is not naturally prepared experimentally. Second, we stress that the qubit adiabatic states $|\psi_\nu(\Phi)\rangle$ are not instantaneous eigenstates, but a perturbative dressing of those in $\hbar\omega_i/\Delta$. We discuss below the consequences of this dressing on the geometric and topological properties of the adiabatic projector.

**Dressed Berry curvature and topology of the family of adiabatic states.**   The ensemble of adiabatic states $|\psi_\nu(\Phi)\rangle$ parametrized by the classical configuration space $\Phi \in [0, 2\pi]^2$ defines a vector bundle. This vector bundle is a smooth deformation of the eigenstates bundle $\left|\psi_\nu^0(\Phi)\right\rangle$ associated to a spectral projector (Fig. 2(a)). As a consequence, the local curvature associated with the adiabatic bundle

$$F_\nu(\Phi) = i\left\langle\partial_{\phi_1}\psi_\nu(\Phi)\middle|\partial_{\phi_2}\psi_\nu(\Phi)\right\rangle - (1 \leftrightarrow 2), \tag{12}$$

differs from the canonical Berry curvature associated with the eigenstates bundle [67]. The curvature (12) is a "dressed Berry curvature" in the sense of a perturbative correction of the Berry curvature of the eigenstates to all orders in $\hbar\omega_i/\Delta$. On the other hand, the Chern number $\mathcal{C}_\nu$ of both bundles are identical. Indeed, the switching on of finite but small frequencies $\omega_1$ and $\omega_2$ is a smooth transformation of the fiber bundle of the eigenstates $\left|\psi_\nu^0(\Phi)\right\rangle$ to that of the adiabatic states $|\psi_\nu(\Phi)\rangle$. Such a smooth transformation does not change the bundle topology.[2]

**Non-adiabatic Landau-Zener transitions.**   The adiabatic projector is defined perturbatively in an adiabatic parameter $\epsilon$, see appendix A. This construction is valid up to non-perturbative effects with typical exponential dependence of the form $\exp(-\alpha/\epsilon)$ [65, 66]. The adiabatic dynamics is the effective dynamics on each subspace, and the non-perturbative transitions between the two subspaces are Landau-Zener transitions. The amplitude of the Landau-Zener transitions can be estimated to obtain the time of validity of the adiabatic approximation [24], $\tau_{\text{adiab}} \approx 0.1\exp(\pi/(4\varepsilon_{\text{adiab}}))T_1$, with $\varepsilon_{\text{adiab}} = \max_{\Phi} \hbar|\left\langle\psi_+^0\middle|\frac{dH}{dt}\middle|\psi_-^0\right\rangle|/(E_+^0 - E_-^0)^2$. In this work, we choose the coupling and the frequencies of the modes such that $\tau_{\text{adiab}} \approx 3100T_1$, allowing to neglect such Landau-Zener transitions. Within this approximation the weight on each adiabatic subspace

$$W_\nu(\Psi) = ||\hat{P}_\nu|\Psi\rangle||^2 = \langle\Psi_\nu|\Psi_\nu\rangle, \tag{13}$$

is a conserved quantity.

## 3.2   Topological splitting of adiabatic components

**Topological splitting.**   The dynamics of each adiabatic component is governed by adiabatic equations of motion containing an anomalous velocity in the direction $n_\perp$ proportional to the curvature $F_\nu(\Phi)$, see appendix B for a detailed derivation.   Similarly than within a hybrid classical-quantum description of a topological pump [15, 24], we assume an incommensurate ratio between the frequencies $\omega_1$ and $\omega_2$ such that a time average reduces by ergodicity to an average over the phases $\Phi \in [0, 2\pi]^2$. The average of the curvature is quantized by the first

---

[2]Note however that this robustness fails for larger frequencies comparable with the spectral gap.

Chern number $\mathcal{C}_\nu$ of the vector bundle of adiabatic states $|\psi_\nu(\boldsymbol{\Phi})\rangle$, and the topological coupling introduced in Sec. 2.1 between the modes and the qubit corresponds to two non-vanishing Chern numbers $\mathcal{C}_+ = -\mathcal{C}_-$. As a result, the topological dynamics splits in energy the two adiabatic components $|\Psi_\pm\rangle$:

$$\langle \hat{n}_\perp \rangle_{\Psi_\pm(t)} = \langle \hat{n}_\perp \rangle_{\Psi_\pm(0)} \mp \frac{|\omega|t}{2\pi}\mathcal{C}_- + \delta n_\perp(t), \tag{14}$$

where $\delta n_\perp(t)$ denotes bounded oscillations, the temporal fluctuations of pumping, discussed in Sec. 4.2.

A cat state is created when the two adiabatic components no longer overlap. After this time of separation, the weight of the state in the region $n_\perp < n_\perp^0$ identifies with the adiabatic weight $W_-(\Psi)$.

**Geometric details of the adiabatic dynamics.** The precise expression of variation of number of quanta $d\langle \hat{n}_i \rangle/dt$ in an adiabatic component is provided in Eq. (B.14) of appendix B.2. This expression differs by two aspects from the transfer of number of quanta obtained within a classical description of the drives (or at least one of them) of a topological pump [15,23,24]. First, due to the quantum nature of the modes, the instantaneous rate is averaged by the phase density $|\chi_\nu(\boldsymbol{\Phi})|^2$ of the adiabatic component. Second, at all orders in adiabatic theory, the equations of motion take a similar form as the first order theory [15,24,35,65], obtained by replacing respectively the eigenenergy and Berry curvature by the adiabatic energy and dressed Berry curvature. The geometric details of the dynamics are affected by perturbative corrections in $\hbar\omega_i/\Delta$, but the above discussed topological drift is not.

Note that by setting a commensurate ratio between the frequencies, the phases follow a closed trajectory on the torus along which the adiabatic curvature can have a non-zero average value even for a trivial Chern number. This would induce a drift at a non-quantized rate depending on the initial phase, referred to as geometric pumping in the Thouless pumps literature [2,68]. Such models were also considered in quantum simulations of spin-orbit-induced anomalous Hall effect using trapped ions [55,56,69].

**Relation with topological pumping.** In relation with the notion of topological pumping, we note that the pumping rate is quantized only for an adiabatic state. For a non-adiabatic state that decomposes into a sum of two adiabatic components of relative weights $W_\pm$, the pumping rate is no longer quantized but rescaled by $(W_- - W_+)$. In the following, we quantify these weights $W_\pm$ for various initial states. We show that a generic initial state is not adiabatic. Therefore, it does not lead to a topological quantized pumping but evolves into a cat state.

## 3.3 Weight of the adiabatic cat state

In this section, we focus on the weight $W_\nu(\Psi)$ (13) of each component of a generic state. We relate them to the quantum geometry of the qubit adiabatic states $|\psi_\nu(\boldsymbol{\Phi})\rangle$, *i.e.* to their dependence on the phases $\boldsymbol{\Phi}$ of the modes, quantified by their quantum metric [34,37]. A topological coupling induces a strong dependence of the adiabatic states on the phases of the modes, such that a generic initial state splits into a linear superposition of adiabatic states with significative weights on each component.

### 3.3.1 General expression of the weights

The phase states in the decomposition (11) being orthogonal, the weight $W_\nu(\Psi)$ reduces to the weight of the wave amplitude $\chi_\nu(\boldsymbol{\Phi})$ defined in (11):

$$W_\nu(\Psi) = \int d^2\boldsymbol{\Phi}\, |\chi_\nu(\boldsymbol{\Phi})|^2. \tag{15}$$

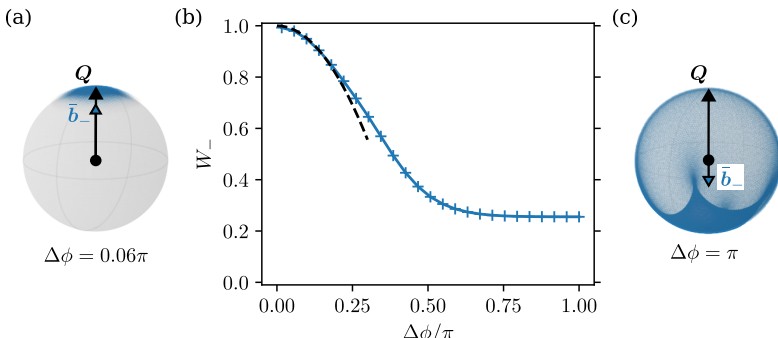

Figure 4: Weight $W_-$ of the ground adiabatic component. The modes are in a Gaussian state centered on $\phi_1^0 = \phi_2^0 = 0$, and we vary their width in phase $\Delta\phi = \Delta\phi_1 = \Delta\phi_2$. The qubit is prepared in $|\uparrow\rangle$ represented by the vector $\mathbf{Q}$ on the Bloch sphere. The phase density of the modes $|\chi(\boldsymbol{\Phi})|^2$ on the torus translates via the map $\boldsymbol{\Phi} \mapsto \mathbf{b}_-(\boldsymbol{\Phi})$ to a density of adiabatic states on the Bloch sphere represented in blue. $\bar{\mathbf{b}}_-$ is the average Bloch vector for this density. The weight of adiabaticity is controlled by the distance of the qubit initial state $\mathbf{Q}$ to the average adiabatic state $\bar{\mathbf{b}}_-$, $W_- = (1 + \bar{\mathbf{b}}_- \cdot \mathbf{Q})/2$. (a) For an initial state localized in phase, $\Delta\phi = 0.06\pi$, the density of adiabatic states is localized on the Bloch sphere such that $\bar{\mathbf{b}}_-$ is near the surface of the sphere. (b) Weight of the cat depending on the width in phase $\Delta\phi$. For small $\Delta\phi$, the weight is controlled by the quantum metric $g_{-,ij}(\boldsymbol{\Phi}^0)$ of the adiabatic states (Eq. (19), dashed line). (c) Limit of large $\Delta\phi$ (quasi Fock state). Due to the topological nature of the coupling, the adiabatic states cover the entire Bloch sphere. This leads to comparable weights on each cat component.

It is instructive to express this weight by representing the qubit states on the Bloch sphere. The qubit adiabatic states $|\psi_\nu(\boldsymbol{\Phi})\rangle$ are represented by a vector $\mathbf{b}_\nu(\boldsymbol{\Phi})$ (see Fig. 2(a)), and the qubit initial state $|\psi_q\rangle$ by $\mathbf{Q}$ (see Fig. 4(a)). The overlap between $|\psi_\nu(\boldsymbol{\Phi})\rangle$ and $|\psi_q\rangle$ is $|\langle\psi_\nu(\boldsymbol{\Phi})|\psi_q\rangle|^2 = (1 + \mathbf{b}_\nu(\boldsymbol{\Phi}) \cdot \mathbf{Q})/2$ such that the weight (15) now reads

$$W_\nu(\Psi) = \frac{1}{2}\left(1 + \bar{\mathbf{b}}_\nu \cdot \mathbf{Q}\right), \tag{16}$$

with the average adiabatic state

$$\bar{\mathbf{b}}_\nu = \int \mathrm{d}^2\boldsymbol{\Phi} \, |\chi(\boldsymbol{\Phi})|^2 \, \mathbf{b}_\nu(\boldsymbol{\Phi}). \tag{17}$$

Let us comment the expression (16). The phase density $|\chi(\boldsymbol{\Phi})|^2$ of the initial state translates via the map $\boldsymbol{\Phi} \mapsto \mathbf{b}_\nu(\boldsymbol{\Phi})$ to a density of adiabatic states on the Bloch sphere, represented in blue on Fig. 4(a) and Fig. 4(c). The average adiabatic state $\bar{\mathbf{b}}_\nu$ is the average Bloch vector for this density, and the weight $W_\nu(\Psi)$ is the distance of the qubit initial state to this average adiabatic state.

In the figure 4(b) we study the effect of the spreading of the initial modes' states on the weights $W_\nu(\Psi)$. We consider the weight on the ground component $W_-(\Psi)$ of an initial state with the modes in a Gaussian state centered on $\phi_1^0 = \phi_2^0 = 0$, and the qubit prepared on $|\uparrow\rangle$ (vector $\mathbf{Q}$ on the north pole). Figure 4(b) displays the evolution of $W_-(\Psi)$ as the width in phase $\Delta\phi = \Delta\phi_1 = \Delta\phi_2$ of the modes is increased. The adiabatic weight is computed numerically from the topological splitting of the adiabatic components discussed in the previous section. For small $\Delta\phi$, the phase density is localized around $\boldsymbol{\Phi}^0$ and $\bar{\mathbf{b}}_\nu \approx \mathbf{b}_\nu(\boldsymbol{\Phi}^0)$ close to the

surface of the Bloch sphere, see Fig. 4(a). As such, if the qubit is prepared with **Q** aligned with $\bar{\mathbf{b}}_-$, as considered on Fig. 4, then $W_- \simeq 1$. This leads to an asymmetric cat state with a small excited component, $W_+ \simeq 0$.

When increasing $\Delta\phi$, the weight of the cat is controlled by the evolution of the qubit adiabatic state $\mathbf{b}_-(\mathbf{\Phi})$ with respect to the phases $\mathbf{\Phi}$. This evolution is encoded into their quantum geometry, which itself is constrained by the topological nature of the coupling. Indeed, a topological coupling corresponds to qubit adiabatic states $\mathbf{b}_-(\mathbf{\Phi})$ covering the entire Bloch sphere when the phases $\mathbf{\Phi}$ vary on $[0, 2\pi]^2$, see Fig. 1(c). As a result, when increasing the width $\Delta\phi$, we average vectors over an increasing support on the Bloch sphere (Fig. 4(c)), reducing the norm $|\bar{\mathbf{b}}_\pm|$ which controls the minimum weight of the cat (Fig. 4(b)). The topological nature of the coupling leads to cat states with comparable weight of each component in the limit of large $\Delta\phi$, *i.e.* in the limit of an initial state with a well-defined number of quanta.

### 3.3.2 Quasi phase states and quantum metric

The only separable states leading to a quantized pumping are those lying in an adiabatic subspace, corresponding either to $W_+ = 1$ or $W_- = 1$. They are pure phase states $\left|\mathbf{\Phi}^0\right\rangle \otimes \left|\psi_\pm(\mathbf{\Phi}^0)\right\rangle$ for which $|\bar{\mathbf{b}}_\pm| = 1$, corresponding to the limit $\Delta\phi_i = 0$ on Fig. 4(b). This limit is recovered for harmonic oscillators in a coherent state with a large average number of quanta $\bar{n}_i$, for which $\Delta\phi_i = 1/\sqrt{\bar{n}_i}$. Given that these states are fully delocalized in quanta number **N**, and thus in energy according to (4), we expect them to be hard to realize. Any other separable state lies at a finite distance from each adiabatic subspace, will not lead to a quantized pumping and will be split into a cat state under time evolution. Let us comment on this adiabatic decomposition for almost pure phase states with small $\Delta\phi$. In this case, the correction to adiabaticity is controlled by the quantum metric $g_{\pm,ij}$ of the adiabatic states [34,37]:

$$g_{\pm,ij} = \text{Re}\langle\partial_{\phi_i}\psi_\pm|(1 - |\psi_\pm\rangle\langle\psi_\pm|)|\partial_{\phi_j}\psi_\pm\rangle. \tag{18}$$

Indeed, the weight (15) is dominated by the local variations of the adiabatic states $|\psi_\pm(\mathbf{\Phi})\rangle$ over the narrow phase support $|\chi(\mathbf{\Phi})|^2$. These variations are encoded by the quantum metric: $|\langle\psi_\pm(\mathbf{\Phi}^0 + \delta\mathbf{\Phi})|\psi_\pm(\mathbf{\Phi}^0)\rangle|^2 = 1 - \sum_{i,j} g_{\pm,ij}(\mathbf{\Phi}^0)\delta\phi_i\delta\phi_j + \mathcal{O}(\delta\phi^3)$. In the limit of a small width $(\Delta\phi_1, \Delta\phi_2)$ the weight (15) for $|\psi_q\rangle = |\psi_\pm(\mathbf{\Phi}^0)\rangle$ reduces to

$$W_\pm(\Psi) = 1 - (\Delta\phi_1)^2 g_{\pm,11}(\mathbf{\Phi}^0) - (\Delta\phi_2)^2 g_{\pm,22}(\mathbf{\Phi}^0) + \mathcal{O}(\Delta\phi^4). \tag{19}$$

Hence for a state close to a phase state, the first correction to $W_\nu$ is quadratic in $\Delta\phi$ with a factor set by the quantum metric of the adiabatic states, as shown in black dashed line on Fig. 4(b).

**Relation with topological pumping** The quasi-phase state limit is relevant in the context of topological pumps, where the phases of the modes are described by classical parameters with a definite value and the fast quantum system is prepared in its corresponding adiabatic state. As discussed above, a non-adiabatic initial state leads to a non-quantized pumping rate rescaled by $(W_- - W_+)$. The result (19) shows that the first correction of the pumping rate due to the quantum fluctuations of the phase is quadratic in $\Delta\phi$ and is controlled by the quantum metric.

Moreover, considering an initial state of the form (11) with the qubit in an eigenstate $\left|\psi_\nu^0(\mathbf{\Phi})\right\rangle$ rather than in an adiabatic state $|\psi_\nu(\mathbf{\Phi})\rangle$ also leads to corrections to the relative weight $(W_- - W_+)$ and to the pumping rate which are quadratic in $\hbar\omega_i/\Delta$. This is similar to the non-adiabatic breaking of topological pumping for an abrupt switching on of the drive of a Thouless pump [36]. See appendix D for quantitative discussions on the difference between eigenstate and adiabatic state. In appendix E, we discuss the role of the qubit initial state on the adiabatic weight, and introduce different types of cat states of equal weights $W_+ = W_- = 1/2$.

# 4 Entanglement and dynamics of cat states

Having characterized the balance between the two components of an adiabatic cat state, we now study the dynamics of each component. We will focus first on the entanglement between the qubit and the two modes, before focusing on their Bloch oscillatory dynamics in the number of quanta representation. We unveil the role of the quantum metric in the entanglement between the qubit and the modes, and show that the topological nature of the coupling induces a strong entanglement.

## 4.1 Entanglement

Adiabatic states naturally entangle the fast qubit with the slow driving modes, a phenomenon out-of-reach of previous Floquet or classical descriptions of the drives [15, 16, 18–22, 24]. We study quantitatively this entanglement as a function of the initial phase spreading $\Delta\phi$ of the modes. We focus on cat states with almost equal weight $W_\nu \simeq 1/2$, that are obtained with $\Phi^0 = 0$ and the qubit prepared in $(|\uparrow\rangle + |\downarrow\rangle)/\sqrt{2}$ following the analysis of section 3.3.

The entanglement between the qubit and the two modes in an adiabatic component $|\Psi_\nu(t)\rangle$, $\nu = \pm$, is captured by the purity $\gamma_\nu(t) = \mathrm{Tr}\left(\rho_{q,\nu}^2(t)\right) = (1 + |\mathbf{Q}_\nu(t)|^2)/2$ of the qubit, where $\rho_{q,\nu}(t)$ is the reduced density matrix of the qubit and $\mathbf{Q}_\nu(t)$ its polarization. From the adiabatic time evolution (Eq. (B.10) in appendix A), we obtain that the qubit is in the statistical mixture of adiabatic states $\mathbf{b}_\nu(\Phi)$ weighted by the translated phase density $|\chi_\nu(\Phi + \omega t)|^2/W_\nu$, i.e.

$$\mathbf{Q}_\nu(t) = \int \mathrm{d}^2\Phi \, \frac{|\chi_\nu(\Phi + \omega t)|^2}{W_\nu} \mathbf{b}_\nu(\Phi).$$ (20)

This is due to the linearity in $\hat{\mathbf{N}}$ of the rotor model which induces an absence of phase dispersion.

### 4.1.1 Entanglement of quasi-phase states

Let us first focus on the adiabatic states with a small $\Delta\phi$, corresponding to quasi-phase states. We show below that for such a state, the entanglement between the qubit and the quantum modes is set by the quantum metric of the adiabatic states.

The translated phase density $|\chi(\Phi + \omega t)|^2$ of the modes is a normalized $2\pi$-periodic Gaussian centered on $\Phi^0 - \omega t$ and of width $(\Delta\phi_1, \Delta\phi_2)$. Plugging into Eq. (20) the expansions of Eqs. (11,15), and $\mathbf{b}_\nu(\Phi)$ in the limit of small $\Delta\phi_1, \Delta\phi_2$, we get

$$\mathbf{Q}_\nu(t) = \mathbf{b}_\nu(\Phi^0 - \omega t) + \frac{1}{2} \sum_i (\Delta\phi_i)^2 \frac{\partial^2 \mathbf{b}_\nu}{\partial \phi_i^2}(\Phi^0 - \omega t) + \mathcal{O}(\Delta\phi^4).$$ (21)

Note that in the limit $\Delta\phi_1 = \Delta\phi_2 = 0$ we recover the classical description of the modes, for which the qubit follows the instantaneous adiabatic state $\mathbf{b}_\nu(\Phi^0 - \omega t)$. From the normalization of the adiabatic states $|\mathbf{b}_\nu|^2 = 1$, we deduce the relation $\mathbf{b}_\nu \cdot \partial_{\phi_i}^2 \mathbf{b}_\nu = -\partial_{\phi_i}\mathbf{b}_\nu \cdot \partial_{\phi_i}\mathbf{b}_\nu = -4g_{\nu,ii}$ where the last equation is an expression of the quantum metric of a two-level system in terms of the Bloch vectors [41, 70]. From this we obtain the expansion at order $\Delta\phi^2$ of the purity

$$\gamma_\nu(t) = 1 - 2(\Delta\phi_1)^2 g_{\nu,11}(\Phi^0 - \omega t) - 2(\Delta\phi_2)^2 g_{\nu,22}(\Phi^0 - \omega t) + \mathcal{O}(\Delta\phi^4).$$ (22)

From the inequality $g_{\nu,11} + g_{\nu,22} \geq |F_\nu|$, originating from the positive semidefiniteness of the quantum geometric tensor [40], we obtain a lower bound on the entanglement between the qubit and the modes. Indeed, for a topological coupling between the qubit and the modes, the average Berry curvature $F_\nu$ is non-vanishing, and set by the Chern number $\mathcal{C}_\nu$, yielding

$$\langle\gamma_\nu\rangle_t \leq 1 - \frac{|\mathcal{C}_\nu|}{\pi}(\Delta\phi)^2 + \mathcal{O}(\Delta\phi^4).$$ (23)

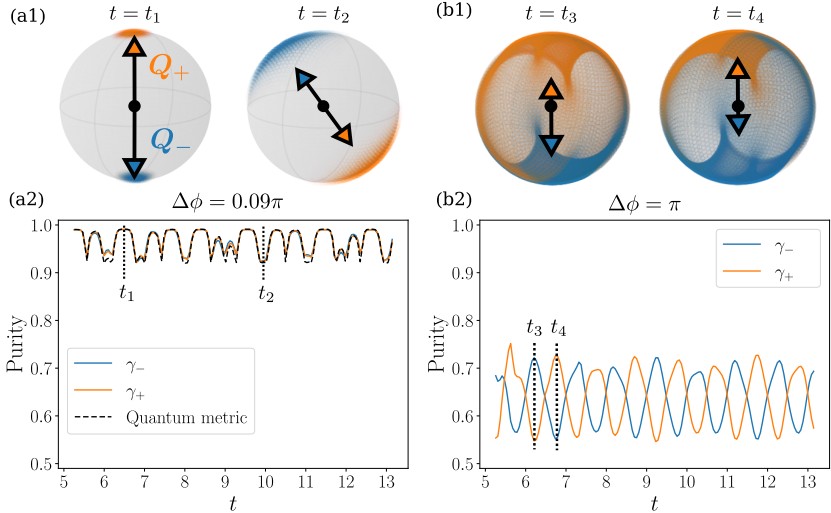

Figure 5: Purity of the qubit $\gamma_\pm(t)$ in each cat components $|\Psi_\pm(t)\rangle$. The qubit is prepared in $(|\uparrow\rangle + |\downarrow\rangle)/\sqrt{2}$. We vary the width in phase $\Delta\phi = \Delta\phi_1 = \Delta\phi_2$ of the initial state. (a1) The phase densities $|\chi_-(\mathbf{\Phi}+\omega t)|^2$ and $|\chi_+(\mathbf{\Phi}+\omega t)|^2$ of each cat components translate to densities of adiabatic states on the Bloch sphere (respectively in blue and orange). The qubit state is the statistical mixture weighted according to these densities with resulting polarizations $\mathbf{Q}_\pm$. For small $\Delta\phi$, the densities of adiabatic states are localized on the Bloch sphere. (a2) Case of small $\Delta\phi$. The adiabatic states cover a larger domain on the Bloch sphere at $t = t_2$ than at $t = t_1$, inducing a larger entanglement $\gamma_-(t_2) < \gamma_-(t_1)$. The temporal variations of the purity are quantified by the quantum metric (Eq. (22), black dashed line). (b1,b2) For large $\Delta\phi$, due to the topological coupling, the adiabatic states cover a large part of the Bloch sphere, corresponding to a high entanglement. At $t = t_3$, the density of excited adiabatic states (in orange) covers a larger support than the density of ground adiabatic states (in blue), leading to a larger entanglement $\gamma_+(t_3) < \gamma_-(t_3)$. The situation is opposite at $t = t_4$.

This demonstrates that a topological pump necessarily entangles the qubit with the modes, a property only captured by the quantum description provided in this paper.

Let us illustrate on a numerical example the role on the purity of the qubit of the geometry of the adiabatic states $\mathbf{b}_\nu(\mathbf{\Phi})$, *i.e.* their dependence on the phases $\mathbf{\Phi}$. The statistical average of the adiabatic states is represented on Fig. 5(a1) for an initial width of the Gaussian state $\Delta\phi = \Delta\phi_1 = \Delta\phi_2 \simeq 0.09\pi$. At a given time $t$, the densities on the torus of each component $|\chi_\pm(\mathbf{\Phi}+\omega t)|^2/W_\nu$ translates into densities of adiabatic states on the Bloch sphere (in blue and orange) encoding the statistical mixture (20) of the qubit in $|\Psi_\pm(t)\rangle$. In Fig. 5(a2), the purity of the qubit after the time of separation is represented respectively in blue and orange for each component. The temporal fluctuations of this purity follow those of the quantum metric represented by a black dashed line, as predicted by Eq. (22). As an illustration, we notice that the quantum metric is smaller at time $t_1$ than at $t_2$, manifesting that the densities of adiabatic states cover a smaller domain of the Bloch sphere at $\mathbf{\Phi} = \mathbf{\Phi}^0 - \omega t_1$ than at $\mathbf{\Phi} = \mathbf{\Phi}^0 - \omega t_2$, as shown on Fig. 5(a1). This translates into a larger purity of the qubit at $t = t_1$ than at $t = t_2$.

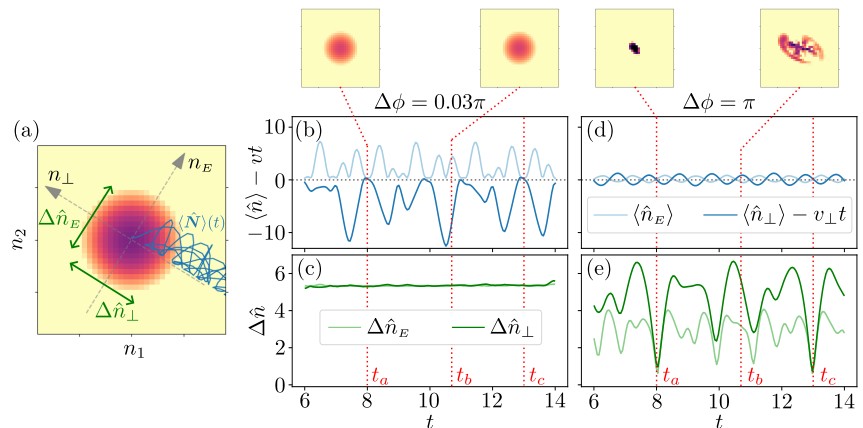

Figure 6: From Bloch oscillations to Bloch breathing. (a) Photon number representation. In blue, sketch of the trajectory of the average number of quanta $\langle \hat{\mathbf{N}} \rangle(t)$ of the ground adiabatic component $|\Psi_-(t)\rangle$ for the example of Fig. 3(a). In green, spreading $\Delta \hat{n}_E$ and $\Delta \hat{n}_\perp$ of $|\Psi_-(t)\rangle$. (b) Bloch oscillations of the center of wavepacket around the topological drift $\mathbf{v}t$ along $n_\perp$, for an initial state localized in phase $\Delta \phi = 0.03\pi$. (c) Time evolution of the spreading for an initial state localized in phase. The state remains Gaussian (insets at $t = t_a$ and $t = t_b$). (d) Initial quasi-Fock state delocalized in phase $\Delta \phi = \pi$. The amplitude of Bloch oscillations around the topological drift are reduced. (e) Bloch breathing. The wavepacket alternates between refocusing ($t = t_a, t_c$) and expansion ($t = t_b$).

### 4.1.2  Entanglement of quasi-Fock states

We now consider adiabatic states with an increasing initial width in phase $\Delta \phi$. Increasing the phase support $\Delta \phi$ increases entanglement: the larger $\Delta \phi$, the larger the support on the Bloch sphere (shown on Fig. 5(b1)), and thus the smaller the polarization (20). The large support on the Bloch sphere originates from the topological nature of the coupling, which imposes that the adiabatic states $\mathbf{b}_\pm(\Phi)$ reach all points of the Bloch sphere as $\Phi$ varies (see Fig. 1(c)). A topologically trivial coupling would lead to a localized distribution of adiabatic states on the Bloch sphere corresponding to an almost pure state of the qubit. This is another manifestation that topological pumping and entanglement between the qubit and the modes are strongly intertwined.

Let us illustrate this increase of entanglement on a numerical example, see Fig. 5(b2). We consider the limit of an initial Fock state entirely delocalized in phase $\Delta \phi = \pi$. In this case, the phase density of the two adiabatic components $|\chi_\pm(\Phi + \omega t)|^2$ have complementary support on the torus $[0, 2\pi]^2$ (see appendix E.2 for details). This translates into two extended supports on the Bloch sphere, represented respectively in blue and orange, inducing a small purity (see Fig. 5(b2) noting that $\gamma = 0.5$ corresponds to a maximal entanglement). At $t = t_3$, the excited density $|\chi_+(\Phi - \omega t_3)|^2$ covers a larger portion of the sphere than the ground density $|\chi_-(\Phi - \omega t_3)|^2$, corresponding to $|\mathbf{Q}_+(t_3)|^2 < |\mathbf{Q}_-(t_3)|^2$ and $\gamma_+(t_3) < \gamma_-(t_3)$ on Fig. 5(b2), while the situation is opposite at $t = t_4$.

### 4.2  Breathing dynamics and Bloch oscillations

We now discuss in more details the oscillations of both the center of mass and of the width in number of each adiabatic component of cat states that manifest themselves on the examples of Fig. 3. This dynamics is reminiscent of Bloch oscillations and Bloch breathing. For a wave-

packet on a lattice submitted to a force, Bloch oscillations correspond to temporal oscillations of the center of the wave-packet while Bloch breathing refers to temporal variations of the width of this wave-packet. The nature of these oscillations and breathing depends on the width of the wavepacket's momentum distribution. Such oscillations were first considered for one dimensional lattices [71–73], latter extended to two dimensions [74–76] and in artificial lattices, see *e.g.* [77,78].

In our context, the lattice sites are indexed by the number of quanta $\mathbf{N} = (n_1, n_2)$ of the modes. The first term of the Hamiltonian (4) is linear in $\mathbf{N}$ and plays the role of the coupling to an electric field $\omega$, while the second term corresponds to a spin-orbit coupling as discussed in section 2.1. Hence, the dynamics of an adiabatic state in $\mathbf{N}$ representation identifies with the Bloch oscillations and breathing of the corresponding wavepacket in the presence of two forces: one longitudinal along the direction $n_E$ of the analogous electric field defined in (8) and one in the transverse direction $n_\perp$, of topological origin, which gives rise to an anomalous transverse velocity. Let us now characterize these Bloch oscillations and breathing in the presence of this transverse topological velocity which, to our knowledge, haven't been discussed yet.

### 4.2.1 Qualitative evolution of an adiabatic component

We sketch on Fig. 6(a) a trajectory of the average number of quanta $\langle \hat{\mathbf{N}} \rangle$ in the ground component $|\Psi_-(t)\rangle$. The two adiabatic subspaces are associated to opposite topological anomalous velocities $\pm\mathbf{v}$, with $\mathbf{v} = (\omega_2, -\omega_1)\mathcal{C}_-/(2\pi)$ the topological velocity of the ground component. This induces a drift of the wavepacket along the direction $n_\perp$ defined in (8), as shown on the figure. In the following, we focus on the dynamics of the component $|\Psi_-(t)\rangle$ around this drift, the results for $|\Psi_+(t)\rangle$ around its opposite drift being similar. We denote as $\Delta\hat{n}_i(t) = [\langle \hat{n}_i^2 \rangle - \langle \hat{n}_i \rangle^2]^{\frac{1}{2}}$ the spreading of $\hat{n}_i$ in $|\Psi_-(t)\rangle$, see Fig. 6(a), $\hat{n}_i$ referring to either $\hat{n}_E$ or $\hat{n}_\perp$. The numbers of quanta $\langle \hat{\mathbf{N}} \rangle$ have temporal fluctuations around the quantized drift $\mathbf{v}t$ represented on Fig. 6(b) for an initial state localized in phase with $\Delta\phi = 0.03\pi$, and on Fig. 6(d) for an initial state delocalized in phase with $\Delta\phi = \pi$.

For an initial state localized in phase, the state remains Gaussian with its initial width $\Delta\hat{n}_i = 1/(2\Delta\phi)$ as illustrated on the insets of Fig. 6(b). When we increase the width $\Delta\phi$ of the initial state, the amplitude of the Bloch oscillations of the center of wavepacket is reduced, while the time variation of the spreading of the wavepacket $\Delta\hat{n}_i$ increases, as shown on Fig. 6(d,e). The Bloch oscillations become Bloch breathing, alternating between refocusing (occuring *e.g.* at $t = t_a$ and $t_c$ on Fig. 6(d,e)) and expansion (*e.g.* at $t = t_b$). The refocusing are discussed in [25] as a photon number boosting mechanism. They occur at quasi-periods detailed in appendix B.4. Let us discuss quantitatively the transition from Bloch oscillations to Bloch breathing using the adiabatic theory developed in Sec. 3.

### 4.2.2 Bloch oscillations of the average number of quanta

In the hybrid classical-quantum description of a topological pump [15,24], the topological quantization of the pumping rate is recovered under a time-average of the instantaneous pumping rate. The temporal fluctuations of this pumping rate originate from inhomogeneities of the energy and Berry curvature. The time average of the Berry curvature is set by the Chern number of the adiabatic states over the torus $[0, 2\pi]^2$. The quantum nature of the mode induces another source of averaging. Let us note $n_i(t, \mathbf{\Phi})$ the classical evolution of the number of quanta obtained in the hybrid description for an initial phase $\mathbf{\Phi}$ of the modes, see Eq. (B.2) in appendix B.3. The average number of quanta $\langle \hat{n}_i \rangle$ in our fully quantum mechanical description corresponds to an average of this classical evolution with respect to the initial phase density $|\chi_\nu(\mathbf{\Phi})|^2$, see Eq. (B.15) in appendix B.3. This induces a quantum average of the Berry curvature, which smoothes out instantaneously the fluctuations of the pumping rate. Thus, an

increase of the support $|\chi_\nu(\mathbf{\Phi})|^2$ reduces the temporal fluctuations of the average number of quanta around its average drift. A topological pumping between quantum modes is indeed "more quantized" than topological pumping between classical modes.

**Relation with Wannier states.**  The reduction of the temporal fluctuations of pumping is set by the width of the phase density $|\chi_\nu(\mathbf{\Phi})|^2$. One would expect that a complete delocalization in phase, $|\chi_\nu(\mathbf{\Phi})|^2 = 1/(2\pi)^2$, averages instantaneously the classical pumping rate over the whole phase space such that $\langle\hat{\mathbf{N}}\rangle(t) = \mathbf{v}t$ without any temporal fluctuations. Such a projected state (11) with uniform delocalization in $\mathbf{\Phi}$ corresponds to a Wannier state for a particle on a lattice, which is topologically obstructed [79–81]. First, let us note that the spreading $\Delta\hat{n}_i$ is infinite in such obstructed Wannier state, making them hard to realize experimentally. Moreover, due to the adiabatic decomposition (10) of the initial (separable) state, such adiabatic states are never realized: the phase density $|\chi_\nu(\mathbf{\Phi})|^2$, defined in (11) contains the density of projection of the qubit initial state $|\langle\psi_\nu(\mathbf{\Phi})|\psi_q\rangle|^2$ which necessarily vanishes on the configuration space for a topological pump, irrespective of $|\psi_q\rangle$. A Wannier state cannot be obtained by the adiabatic decomposition of a separable state, and the temporal fluctuations of the pumping remain finite.

### 4.2.3  Bloch breathing of the spreading

We now express the time evolution of the spreading $\Delta n_i$ of the number of quanta in terms of the adiabatic trajectories, to explain the transition between Bloch oscillations and Bloch breathing. When the phase density $|\chi_\nu(\mathbf{\Phi})|^2$ is localized around $\mathbf{\Phi}^0$, the center of mass performs Bloch oscillations following the classical trajectory $n_i(t, \mathbf{\Phi}^0)$.  When the phase support $|\chi_\nu(\mathbf{\Phi})|^2$ increases, we sum the contribution of different classical trajectories $n_i(t, \mathbf{\Phi})$ that spread around $n_i(t, \mathbf{\Phi}^0)$. This spreading of the wavepacket is captured by the variance $\mathrm{Var}[n_i(t, \mathbf{\Phi})]$ of the classical trajectories with respect to the initial phase distribution $|\chi_\nu(\mathbf{\Phi})|^2$, see Eq. (B.23). We show in appendix B.3 that the time evolution of the spreading $\Delta\hat{n}_i$ in the adiabatic state $|\Psi_\nu(t)\rangle$ is indeed related to this variance, and contains additional terms involving the quantum metric (see Eq. (B.24)).

For a quasi-phase state with a narrow distribution $|\chi_\nu(\mathbf{\Phi})|^2$, the classical trajectories with an initial phase on this support are all similar, and the variance vanishes.  As a result, the quantum fluctuations do not vary in time $\Delta\hat{n}_i(t) \simeq \Delta\hat{n}_i(0) \simeq 1/(2\Delta\phi)$, as shown in Fig. 6(c).  In the case of large $\Delta\phi$, the classical trajectories start to significantly spread for different initial phases, leading to a large $\mathrm{Var}[n_i(t, \mathbf{\Phi})]$ and an expansion of the wavepacket, seen for example at $t = t_b$ on Fig. 6(e). At quasi-periods $T$ discussed in appendix B.3, the classical trajectories lead almost to the same quantized drift $\mathbf{N}(T, \mathbf{\Phi}) \simeq \mathbf{v}T$ for all initial phases $\mathbf{\Phi}$, such that $\mathrm{Var}[n_i(T, \mathbf{\Phi})] \simeq 0$. The wave-packet refocuses at these quasi-periods, seen at $t = t_a$ and $t = t_c$ on Fig. 6(e).

**Effect of the adiabatic projection on the refocusing.**  The lack of perfect refocusing is usually discussed in the literature as a lack of rephasing $\mathrm{Var}[n_i(T, \mathbf{\Phi})] \gtrsim 0$ [25, 74], such that $(\Delta\hat{n}_i)(T) \gtrsim (\Delta\hat{n}_i)(t = 0)$.  We note another important point about this refocusing:  it does not correspond to a refocusing of the initial state $|\Psi(t = 0)\rangle$ but of its adiabatic projection $|\Psi_\nu(t = 0)\rangle$. However, the adiabatic projection affects the spread of the wavepacket, such that the adiabatic cat component do not refocus with the same spread as in the initial state. This is visible on Fig. 6(e): even though we consider an initial Fock state, the adiabatic component does not refocus into a Fock state at $t = t_a$. Indeed, as discussed above the phase distribution $|\chi_\nu(\mathbf{\Phi})|^2$ is not fully delocalized on the torus, such that by Heisenberg inequality the distribution of number of quanta in $|\Psi_\nu(t = 0)\rangle$ is not fully localized. For the cat state with equal weight, we discuss in appendix E.2 that $|\chi_\nu(\mathbf{\Phi})|^2$ covers approximately half the torus, corresponding to a spread of order $\pi/4$, such that $\Delta\hat{n}_i(t = 0) \geq 2/\pi \simeq 0.63$. This is approximately the values of the spreading at the refocusing times $t = t_a, t_c$ on Fig. 6(e).

# 5 Conclusion

In this work, we have shown that the dynamics of a qubit coupled topologically to two slow quantum modes generically creates a cat state, a superposition of two adiabatic states with mesoscopically distinct energy content. We developed an adiabatic approximation method which shows that the topological splitting of the two cat components at a quantized rate is robust at all orders in the ratio between the modes' frequency and the Bohr frequency of the qubit. In contrast, topological pumping at a quantized rate requires to prepare the initial state in a fine-tuned adiabatic state. For each adiabatic component of the cat, the topological nature of the coupling induces an intrinsic entanglement between the qubit and the modes. We unveil topological constraints on this entanglement by relating it to the quantum geometry and quantum metric of the adiabatic states.

Our results also describe the effect of the quantum nature of the drive of a Thouless pump. We showed that the splitting of an initial wavepacket into a cat state is generic and responsible for the breakdown of the quantization of pumping on short timescales. For a sudden switch on of the drive in a coherent state, the deviation from quantization contains a quadratic contribution in the quantum fluctuation of the driving phase which adds up to the known quadratic contribution in the drive frequency [36]. Nonetheless, each cat component drifts at a quantized rate at all orders in the drive frequency. As such, when the two components no longer overlap, we can prepare an adiabatic wavepacket of particle of quantized average drift using a projective measurement. Our analysis also shows that the quantum fluctuations of the driving phase stabilizes the pump: it reduces the time variations of the center-of-mass around the quantized drift.

The realization of such topological adiabatic cat states opens interesting perspectives, in particular to elaborate protocols to disentangle the qubit from the quantum modes, creating an entangled cat state between the modes. One can build on existing protocols for a superconducting qubit dispersively coupled to quantum cavities, with cat composed of coherent states non-entangled with the qubit, of typical form $(|\alpha_1, \alpha_2, \uparrow\rangle + |\beta_1, \beta_2, \downarrow\rangle)/\sqrt{2}$ [82]. From our analysis of Sec. 4.1, similar states are obtained from an adiabatic cat state in the quasi-phase limit at a time $t$ where a small value of the quantum metric $g_{ij}(\Phi^0 - \omega t)$ is reached. Besides, it is worth pointing out that the topological splitting of the two adiabatic components allows for the experimental preparation of adiabatic states, by using a projection on the number of quanta $(n_1, n_2)$ such that $n_1 - n_2 > n_1^0 - n_2^0$ after the time of separation. In the perspective of a superconducting qubit coupled to quantum cavities, such a measurement protocol can be adapted from the methods of photon number resolution [83] using an additional qubit dispersively coupled to the two cavities.

Finally, let us stress that we have focused on the adiabatic limit of a quantum description of a Floquet system. We characterized the entanglement between the drives and the driven quantum system in terms of the quantum geometry of adiabatic states. Extending this relation between entanglement and geometry beyond the adiabatic limit [84, 85] is a natural and stimulating perspective.

## Acknowledgments

We are grateful to Audrey Bienfait, Emmanuel Flurin, Benjamin Huard and Zaki Leghtas for insightful discussions.

# A  Adiabatic projector

## A.1  Time evolution of phase states

We determine the time evolution of a phase eigenstate $|\Psi(t=0)\rangle = |\mathbf{\Phi}\rangle \otimes |\psi\rangle$ with $|\psi\rangle$ is an arbitrary state of the two-level system. We consider the Hamiltonian of the rotor model $\hat{H}_{\text{tot}} = \hbar\omega \cdot \hat{\mathbf{N}} + H(\hat{\mathbf{\Phi}})$ where $\omega = (\omega_1, \cdots, \omega_N)$, $\hat{\mathbf{N}} = (\hat{n}_1, \cdots, \hat{n}_N)$, $\hat{\mathbf{\Phi}} = (\hat{\phi}_1, \cdots, \hat{\phi}_N)$, where the operators $\hat{n}_i$ and $\hat{\phi}_i$ are conjugated $[\hat{n}_i, \hat{\phi}_j] = \mathrm{i}\delta_{i,j}\mathbf{1}$, and $\omega \cdot \hat{\mathbf{N}} = \sum_i \omega_i \hat{n}_i$. In the interaction representation with respect to the Hamiltonian of the modes, the time evolved state is:

$$|\Psi_I(t)\rangle = \exp\left[\mathrm{i}t\omega \cdot \hat{\mathbf{N}}\right]|\Psi(t)\rangle . \tag{A.1}$$

The dynamics of $|\Psi_I(t)\rangle$ is governed by the Hamiltonian in the interaction representation:

$$\hat{H}_I(t) = \exp\left[\mathrm{i}t\omega \cdot \hat{\mathbf{N}}\right]H(\hat{\mathbf{\Phi}})\exp\left[-\mathrm{i}t\omega \cdot \hat{\mathbf{N}}\right] \tag{A.2}$$

$$= H(\hat{\mathbf{\Phi}} - \omega t), \tag{A.3}$$

since the operators $\hat{n}_i$ are generators of the phase translation. As a consequence,

$$\hat{H}_I(t)(|\mathbf{\Phi}\rangle \otimes |\psi\rangle) = |\mathbf{\Phi}\rangle \otimes H(\mathbf{\Phi} - \omega t)|\psi\rangle , \tag{A.4}$$

such that the time-evolution of the initial state $|\mathbf{\Phi}\rangle \otimes |\psi\rangle$ in the interaction representation is

$$|\Psi_I(t)\rangle = \mathcal{T}\exp\left[-\frac{\mathrm{i}}{\hbar}\int_0^t \mathrm{d}\tau \, \hat{H}_I(\tau)\right]|\mathbf{\Phi}\rangle \otimes |\psi\rangle$$

$$= |\mathbf{\Phi}\rangle \otimes U(t; \mathbf{\Phi})|\psi\rangle , \tag{A.5}$$

where $\mathcal{T}$ denotes time-ordering and $U(t; \mathbf{\Phi}) = \mathcal{T}\exp\left[-\frac{\mathrm{i}}{\hbar}\int_0^t \mathrm{d}\tau \, H(\mathbf{\Phi} - \omega\tau)\right]$ is the time evolution operator associated to the time-dependent Hamiltonian $H(\mathbf{\Phi} - \omega t)$ for classical modes. We then obtain the time-evolved state in the Schrödinger representation

$$|\Psi(t)\rangle = \exp\left[-\mathrm{i}t\omega \cdot \hat{\mathbf{N}}\right]|\Psi_I(t)\rangle \tag{A.6}$$

$$= |\mathbf{\Phi} - \omega t\rangle \otimes U(t; \mathbf{\Phi})|\psi\rangle . \tag{A.7}$$

## A.2  Construction of the adiabatic states

We construct the adiabatic states of the two-level system $|\psi_\nu(\mathbf{\Phi})\rangle$ such that the family of states $|\mathbf{\Phi}\rangle \otimes |\psi_\nu(\mathbf{\Phi})\rangle$ with $\mathbf{\Phi} \in [0, 2\pi]^2$ is stable under the dynamics governed by the total Hamiltonian $H_{\text{tot}} = \sum_i \hbar\omega_i \hat{n}_i + H(\hat{\mathbf{\Phi}})$. This family of states corresponds to the image of the associated adiabatic projector

$$\hat{P}_\nu = \int \mathrm{d}\mathbf{\Phi} \, |\mathbf{\Phi}\rangle \langle\mathbf{\Phi}| \otimes |\psi_\nu(\mathbf{\Phi})\rangle \langle\psi_\nu(\mathbf{\Phi})| \tag{A.8}$$

$$= \int \mathrm{d}\mathbf{\Phi} \, |\mathbf{\Phi}\rangle \langle\mathbf{\Phi}| \otimes \pi_\nu(\mathbf{\Phi}). \tag{A.9}$$

We first construct the family of adiabatic projectors of the two-level system $\pi_\nu(\mathbf{\Phi}) = |\psi_\nu(\mathbf{\Phi})\rangle \langle\psi_\nu(\mathbf{\Phi})|$. As detailed in appendix A.1, for an arbitrary state $|\psi\rangle$ of the fast quantum degree of freedom, the phase eigenstates evolve according to

$$\exp(-\mathrm{i}\hat{H}_{\text{tot}}t/\hbar)(|\mathbf{\Phi}\rangle \otimes |\psi\rangle) = |\mathbf{\Phi} - \omega t\rangle \otimes U(t; \mathbf{\Phi})|\psi\rangle , \tag{A.10}$$

with $U(t; \mathbf{\Phi})$ the time evolution operator associated to the Floquet Hamiltonian $H(\mathbf{\Phi} - \omega t)$. Thus the previous family of states is stable if the projectors $\pi_\nu(\mathbf{\Phi})$ satisfy

$$U(t; \mathbf{\Phi})\pi_\nu(\mathbf{\Phi})U(t; \mathbf{\Phi})^\dagger = \pi_\nu(\mathbf{\Phi} - \omega t), \tag{A.11}$$

or equivalently $-i\hbar\omega \cdot \nabla_{\boldsymbol{\Phi}} \pi_{\nu}(\boldsymbol{\Phi}) = [H(\boldsymbol{\Phi}), \pi_{\nu}(\boldsymbol{\Phi})]$. This equation can be solved perturbatively, assuming that the eigenstates evolve slowly. Similarly to [86], we rescale all the frequencies by a dimensionless parameter $\epsilon$: $\omega \rightarrow \epsilon\omega$. We search a projector $\pi_{\nu}(\boldsymbol{\Phi})$ expressed a formal series of $\epsilon$

$$\pi_{\nu}(\boldsymbol{\Phi}) = \sum_k \epsilon^k \pi_{\nu,k}(\boldsymbol{\Phi}) = \pi_{\nu,0}(\boldsymbol{\Phi}) + \epsilon\pi_{\nu,1}(\boldsymbol{\Phi}) + \dots, \tag{A.12}$$

solution of the equations

$$-i\epsilon\hbar\omega \cdot \nabla_{\boldsymbol{\Phi}} \pi_{\nu}(\boldsymbol{\Phi}) = [H(\boldsymbol{\Phi}), \pi_{\nu}(\boldsymbol{\Phi})], \tag{A.13}$$

$$\pi_{\nu}(\boldsymbol{\Phi})^2 = \pi_{\nu}(\boldsymbol{\Phi}). \tag{A.14}$$

Such a solution exists as an asymptotic series in $\epsilon$ [65, 66, 87].

We detail the determination of the two first terms. The conditions (A.13) and (A.14) give for the order 0

$$[H(\boldsymbol{\Phi}), \pi_{\nu,0}(\boldsymbol{\Phi})] = 0, \tag{A.15}$$

$$\pi_{\nu,0}(\boldsymbol{\Phi})^2 = \pi_{\nu,0}(\boldsymbol{\Phi}), \tag{A.16}$$

which are satisfied for a family of projectors $\pi_{\nu,0}(\boldsymbol{\Phi}) = \left|\psi_{\nu}^0(\boldsymbol{\Phi})\right\rangle\left\langle\psi_{\nu}^0(\boldsymbol{\Phi})\right|$ on eigenstates $\left|\psi_{\nu}^0(\boldsymbol{\Phi})\right\rangle$ of $H(\boldsymbol{\Phi})$ associated to the eigen-energy $E_{\nu}^0(\boldsymbol{\Phi})$, $H(\boldsymbol{\Phi})\left|\psi_{\nu}^0(\boldsymbol{\Phi})\right\rangle = E_{\nu}^0(\boldsymbol{\Phi})\left|\psi_{\nu}^0(\boldsymbol{\Phi})\right\rangle$. At order 1, the conditions (A.13) and (A.14) read

$$[H(\boldsymbol{\Phi}), \pi_{\nu,1}(\boldsymbol{\Phi})] = i\hbar\omega \cdot \nabla_{\boldsymbol{\Phi}} \pi_{\nu,0}(\boldsymbol{\Phi}), \tag{A.17}$$

$$\pi_{\nu,1}(\boldsymbol{\Phi}) = \pi_{\nu,1}(\boldsymbol{\Phi})\pi_{\nu,0}(\boldsymbol{\Phi}) + \pi_{\nu,0}(\boldsymbol{\Phi})\pi_{\nu,1}(\boldsymbol{\Phi}). \tag{A.18}$$

When the eigenstate $\left|\psi_{\nu}^0(\boldsymbol{\Phi})\right\rangle$ is non-degenerate for all $\boldsymbol{\Phi} \in [0, 2\pi]^2$, these equations are satisfied for

$$\pi_{\nu,1}(\boldsymbol{\Phi}) = \sum_{\mu \neq \nu} \left|\psi_{\mu}^0(\boldsymbol{\Phi})\right\rangle \frac{\sum_i \hbar\omega_i A_{\mu\nu,i}^0(\boldsymbol{\Phi})}{E_{\mu}^0(\boldsymbol{\Phi}) - E_{\nu}^0(\boldsymbol{\Phi})} \left\langle\psi_{\nu}^0(\boldsymbol{\Phi})\right| + \text{h.c.}, \tag{A.19}$$

with $A_{\mu\nu,i}^0(\boldsymbol{\Phi}) = i\left\langle\psi_{\mu}^0(\boldsymbol{\Phi})\middle|\partial_{\phi_i}\psi_{\nu}^0(\boldsymbol{\Phi})\right\rangle$ the components of the non-Abelian Berry connection of the eigenstates. Thus the adiabatic state at order 1 decomposes on the eigenstates similarly to the usual time dependent perturbation theory

$$\left|\psi_{\nu}(\boldsymbol{\Phi})\right\rangle = \left|\psi_{\nu}^0(\boldsymbol{\Phi})\right\rangle + \epsilon \sum_{\mu \neq \nu} \left(\frac{\sum_i \hbar\omega_i A_{\mu\nu,i}^0(\boldsymbol{\Phi})}{E_{\mu}^0(\boldsymbol{\Phi}) - E_{\nu}^0(\boldsymbol{\Phi})}\right)\left|\psi_{\mu}^0(\boldsymbol{\Phi})\right\rangle + \mathcal{O}(\epsilon^2). \tag{A.20}$$

## B  Adiabatic dynamics

We derive the time-evolution of each adiabatic component (11) of the initial state

$$\left|\Psi_{\nu}(t=0)\right\rangle = \int \mathrm{d}^2\boldsymbol{\Phi} \; \chi_{\nu}(\boldsymbol{\Phi})\left|\boldsymbol{\Phi}\right\rangle \otimes \left|\psi_{\nu}(\boldsymbol{\Phi})\right\rangle. \tag{B.1}$$

We first derive the time evolution of each phase component $\left|\boldsymbol{\Phi}\right\rangle \otimes \left|\psi_{\nu}(\boldsymbol{\Phi})\right\rangle$. Then we derive the evolution of the average number of quanta, and of the spreading of the number of quanta. We show that due the linearity in $\hat{\mathbf{N}}$ of the rotor Hamiltonian we can express these observables from the classical trajectories $n_i(t, \boldsymbol{\Phi})$ [24] which are obtained in an hybrid classical-quantum description of the qubit-mode coupling

$$n_i(t, \boldsymbol{\Phi}) = \int_0^t \mathrm{d}t' \left(\frac{1}{\hbar}\frac{\partial E_{\nu}}{\partial \phi_i}(\boldsymbol{\Phi} - \omega t') + \sum_j \omega_j F_{\nu,ij}(\boldsymbol{\Phi} - \omega t')\right), \tag{B.2}$$

with the adiabatic energy function

$$E_\nu(\boldsymbol{\Phi}) = \langle \psi_\nu(\boldsymbol{\Phi})| H(\boldsymbol{\Phi}) |\psi_\nu(\boldsymbol{\Phi})\rangle \,, \tag{B.3}$$

and dressed Berry curvature

$$F_{\nu,ij}(\boldsymbol{\Phi}) = i \left\langle \partial_{\phi_i}\psi_\nu(\boldsymbol{\Phi}) \middle| \partial_{\phi_j}\psi_\nu(\boldsymbol{\Phi}) \right\rangle - (i \leftrightarrow j) \,. \tag{B.4}$$

### B.1 Time evolution of the qubit adiabatic states

In appendix A.2 we constructed the projectors $\pi_\nu(\boldsymbol{\Phi}) = |\psi_\nu(\boldsymbol{\Phi})\rangle \langle \psi_\nu(\boldsymbol{\Phi})|$ on the states of the two-level system such that the image of the adiabatic projector $\hat{P}_\nu$ is stable under the dynamics. As discussed in appendix A.2, such states $|\psi_\nu(\boldsymbol{\Phi})\rangle$ have to satisfy

$$U(t;\boldsymbol{\Phi}) |\psi_\nu(\boldsymbol{\Phi})\rangle = e^{i\theta_\nu(t;\boldsymbol{\Phi})} |\psi_\nu(\boldsymbol{\Phi} - \omega t)\rangle \,, \tag{B.5}$$

with $U(t;\boldsymbol{\Phi})$ the time evolution operator associated to the time-dependent Hamiltonian $H(\boldsymbol{\Phi} - \omega t)$ and with $\theta_\nu(t;\boldsymbol{\Phi})$ a phase factor. We show that this phase factor is related to the Aharonov-Anandan phase. By definition, the time evolution operator $U(t;\boldsymbol{\Phi})$ satisfies

$$i\hbar \frac{\mathrm{d}}{\mathrm{d}t} U(t;\boldsymbol{\Phi}) = H(\boldsymbol{\Phi} - \omega t)U(t;\boldsymbol{\Phi}), \tag{B.6}$$

such that (B.5) gives

$$\frac{1}{i\hbar}H(\boldsymbol{\Phi} - \omega t)|\psi_\nu(\boldsymbol{\Phi} - \omega t)\rangle = i\frac{\partial \theta_\nu}{\partial t}(t;\boldsymbol{\Phi})|\psi_\nu(\boldsymbol{\Phi} - \omega t)\rangle - \sum_i \omega_i \left| \partial_{\phi_i}\psi_\nu(\boldsymbol{\Phi} - \omega t)\right\rangle \,. \tag{B.7}$$

The phase factor $\theta_\nu(t;\boldsymbol{\Phi})$ is then given by

$$\theta_\nu(t;\boldsymbol{\Phi}) = \int_0^t \mathrm{d}t' \left( -\frac{1}{\hbar}E_\nu(\boldsymbol{\Phi} - \omega t') - \sum_i \omega_i A_{\nu,i}(\boldsymbol{\Phi} - \omega t') \right) \,, \tag{B.8}$$

with the energy function $E_\nu$ (B.3) and the (generalized) Berry connection

$$A_{\nu,i}(\boldsymbol{\Phi}) = i \left\langle \psi_\nu(\boldsymbol{\Phi}) \middle| \partial_{\phi_i}\psi_\nu(\boldsymbol{\Phi}) \right\rangle \,. \tag{B.9}$$

### B.2 Time evolution of the average number of quanta

From appendix B.1, the time evolution of the projected state (A.20) reads

$$|\Psi_\nu(t)\rangle = \int \mathrm{d}^2\boldsymbol{\Phi}\, \chi_\nu(\boldsymbol{\Phi})e^{i\theta_\nu(t;\boldsymbol{\Phi})} |\boldsymbol{\Phi} - \omega t\rangle \otimes |\psi_\nu(\boldsymbol{\Phi} - \omega t)\rangle \,, \tag{B.10}$$

with the phase factor $\theta_\nu(t;\boldsymbol{\Phi})$ given by (B.8). The Ehrenfest theorem reads

$$\frac{\mathrm{d}}{\mathrm{d}t}\langle \hat{n}_i \rangle_{\Psi_\nu(t)} = \frac{1}{i\hbar}\langle [\hat{n}_i, \hat{H}_{\mathrm{tot}}]\rangle_{\Psi_\nu(t)} \,, \tag{B.11}$$

where from the Hamiltonian (4) we have $[\hat{n}_i, \hat{H}_{\mathrm{tot}}] = i\frac{\partial H}{\partial \phi_i}(\hat{\boldsymbol{\Phi}})$ such that

$$\frac{\mathrm{d}}{\mathrm{d}t}\langle \hat{n}_i \rangle_{\Psi_\nu(t)} = \int \mathrm{d}^2\boldsymbol{\Phi}\, \frac{|\chi_\nu(\boldsymbol{\Phi})|^2}{W_\nu}\frac{1}{\hbar}\langle \psi_\nu(\boldsymbol{\Phi} - \omega t)| \frac{\partial H}{\partial \phi_i}(\boldsymbol{\Phi} - \omega t)|\psi_\nu(\boldsymbol{\Phi} - \omega t)\rangle \,, \tag{B.12}$$

with the normalization factor $W_\nu = \langle \Psi_\nu(t)|\Psi_\nu(t)\rangle$. The average value of the derivative of the Hamiltonian can be written

$$\langle \psi_\nu| \frac{\partial H}{\partial \phi_i} |\psi_\nu\rangle = \frac{\partial}{\partial \phi_i} (\langle \psi_\nu| H |\psi_\nu\rangle) - \langle \partial_{\phi_i}\psi_\nu | H |\psi_\nu\rangle - \langle \psi_\nu| H |\partial_{\phi_i}\psi_\nu\rangle, \tag{B.13}$$

where the dependence on $\Phi - \omega t$ is implicit. The first term of this equation gives the term of variation of energy $E_\nu(\Phi)$ (B.3). Using the expression (B.7) for $H|\psi_\nu\rangle$ and using the normalization condition $\langle \partial_{\phi_i}\psi_\nu | \psi_\nu\rangle = -\langle \psi_\nu | \partial_{\phi_i}\psi_\nu\rangle$, we write the last two terms of (B.13) in terms of the curvature $F_{\nu,ij}(\Phi)$ (B.4) such that we obtain the expression of the pumping rate

$$\frac{d}{dt}\langle \hat{n}_i\rangle_{\Psi_\nu(t)} = \int d^2\Phi \, \frac{|\chi_\nu(\Phi)|^2}{W_\nu} \left( \frac{1}{\hbar}\frac{\partial E_\nu}{\partial \phi_i}(\Phi - \omega t) + \sum_{j\neq i} \omega_j F_{\nu,ij}(\Phi - \omega t) \right). \tag{B.14}$$

As a result, the time evolution of the average number of quanta reduces to a statistical average of the classical trajectories with respect to the (normalized) initial phase density $|\chi_\nu(\Phi)|^2/W_\nu$

$$\langle \hat{n}_i\rangle_{\Psi_\nu(t)} = \langle \hat{n}_i\rangle_{\Psi_\nu(t=0)} + \int d^2\Phi \, \frac{|\chi_\nu(\Phi)|^2}{W_\nu} n_i(t,\Phi). \tag{B.15}$$

### B.3 Time evolution of the quantum fluctuations

We derive the time evolution of the quantum fluctuation, or spreading, of the modes' number of quanta in an adiabatic component $|\Psi_\nu(t)\rangle$ (11)

$$[\Delta \hat{n}_i(t)]^2 = \langle \hat{n}_i^2\rangle_{\Psi_\nu(t)} - \langle \hat{n}_i\rangle_{\Psi_\nu(t)}^2. \tag{B.16}$$

with $\langle \hat{O}\rangle_{\Psi_\nu(t)} = \langle \Psi_\nu(t)| \hat{O} |\Psi_\nu(t)\rangle / \langle \Psi_\nu(t)|\Psi_\nu(t)\rangle$. We note

$$\xi_\nu(t,\Phi) = \chi_\nu(\Phi)e^{i\theta_\nu(t;\Phi)}/W_\nu, \tag{B.17}$$

the normalized wavefunction entering the time-evolved state (B.10). Using $\langle \phi_i'| \hat{n}_i |\phi_i\rangle = -i\partial_{\phi_i}\delta(\phi_i' - \phi_i)$, we get after a few lines

$$\langle \hat{n}_i^2\rangle_{\Psi_\nu(t)} = \int d\Phi \, \xi_\nu(t,\Phi)^* \left( i\frac{\partial}{\partial \phi_i} + A_{\nu,i}(\Phi - \omega t) \right)^2 \xi_\nu(t,\Phi)$$
$$+ \int d^2\Phi \, |\xi_\nu(t,\Phi)|^2 g_{\nu,ii}(\Phi - \omega t), \tag{B.18}$$

with $g_{\nu,ii}$ the quantum metric of the adiabatic states (18). Let us comment this equation. In a single band approximation of Bloch oscillations, we ignore the rotation of the states $|\psi_\nu(\Phi)\rangle$ such that we assume that $\xi_\nu(t,\Phi)$ is the wavefunction of the modes ignoring the role of the projection $\hat{P}_\nu$. The average value of $\hat{n}_i^2$ is then given by (B.18) without the connection $A_{\nu,i}$ and the metric $g_{\nu,ii}$. Here the first line corresponds to the average value of the projected observable $(\hat{P}_\nu \hat{n}_i \hat{P}_\nu)^2$, where $\hat{P}_\nu \hat{n}_i \hat{P}_\nu$ reduces to a covariant derivative with the connection $A_{\nu,i}$ in the representation $|\Phi\rangle \otimes |\psi_\nu(\Phi)\rangle$. The second term involving the quantum metric originates from the difference between the observables and the projected observables

$$\hat{P}_\nu \hat{n}_i^2 \hat{P}_\nu = (\hat{P}_\nu \hat{n}_i \hat{P}_\nu)^2 + \hat{P}_\nu \hat{n}_i(1 - \hat{P}_\nu)\hat{n}_i \hat{P}_\nu, \tag{B.19}$$

such that we show $\langle \hat{P}_\nu \hat{n}_i(1 - \hat{P}_\nu)\hat{n}_i \hat{P}_\nu\rangle = \int d^2\Phi |\xi_\nu(t,\Phi)|^2 g_{\nu,ii}(\Phi - \omega t)$.

Let us express the time evolution (B.18) it terms of the classical trajectories $n_i(t,\Phi)$ (B.2). Using $A_{\nu,i}(\Phi - \omega t) - A_{\nu,i}(\Phi) = \int_0^t dt' \sum_j \omega_j \partial_j A_{\nu,j}(\Phi - \omega t')$ and $F_{\nu,ij} = \partial_i A_{\nu,j} - \partial_j A_{\nu,i}$, these trajectories can be expressed in terms of the phase factor $\theta_\nu(t,\Phi)$ (B.8) as

$$n_i(t,\Phi) = -\frac{\partial \theta_\nu}{\partial \phi_i}(\Phi;t) + A_{\nu,i}(\Phi - \omega t) - A_{\nu,i}(\Phi). \tag{B.20}$$

After developing the time evolution (B.17) of the wavefunction $\xi_\nu(t, \mathbf{\Phi})$ we obtain

$$\langle \hat{n}_i^2 \rangle_{\Psi_\nu(t)} = \int \frac{\mathrm{d}^2\mathbf{\Phi}}{W_\nu} \chi_\nu(\mathbf{\Phi})^* \left( i \frac{\partial}{\partial \phi_i} + A_{\nu,i}(\mathbf{\Phi}) \right)^2 \chi_\nu(\mathbf{\Phi}) + \int \frac{\mathrm{d}^2\mathbf{\Phi}}{W_\nu} |\chi_\nu(\mathbf{\Phi})|^2 n_i(t, \mathbf{\Phi})$$

$$+ 2 \int \frac{\mathrm{d}^2\mathbf{\Phi}}{W_\nu} J_{\nu,i}(\mathbf{\Phi}) n_i(t, \mathbf{\Phi}) + \int \frac{\mathrm{d}^2\mathbf{\Phi}}{W_\nu} |\chi_\nu(\mathbf{\Phi})|^2 g_{\nu,ii}(\mathbf{\Phi} - \omega t)$$

$$= \langle \hat{n}_i^2 \rangle_{\Psi_\nu(t=0)} + \int \frac{\mathrm{d}^2\mathbf{\Phi}}{W_\nu} |\chi_\nu(\mathbf{\Phi})|^2 n_i(t, \mathbf{\Phi}) + 2 \int \frac{\mathrm{d}^2\mathbf{\Phi}}{W_\nu} J_{\nu,i}(\mathbf{\Phi}) n_i(t, \mathbf{\Phi})$$

$$+ \int \frac{\mathrm{d}^2\mathbf{\Phi}}{W_\nu} |\chi_\nu(\mathbf{\Phi})|^2 (g_{\nu,ii}(\mathbf{\Phi} - \omega t) - g_{\nu,ii}(\mathbf{\Phi})), \tag{B.21}$$

with the current density of the initial state

$$J_{\nu,i} = \frac{i}{2} \left( \chi_\nu^* \frac{\partial \chi_\nu}{\partial \phi_i} - \chi_\nu \frac{\partial \chi_\nu^*}{\partial \phi_i} \right) + |\chi_\nu|^2 A_{\nu,i}, \tag{B.22}$$

satisfying $\langle \hat{n}_i \rangle_{\Psi_\nu(t=0)} = \int J_{\nu,i}(\mathbf{\Phi}) \mathrm{d}\mathbf{\Phi} / W_\nu$.

As a result, the time evolution of the spreading is given by the variance

$$\mathrm{Var}_{|\chi_\nu|^2}[n_i(t, \mathbf{\Phi})] = \int \mathrm{d}^2\mathbf{\Phi} \frac{|\chi_\nu(\mathbf{\Phi})|^2}{W_\nu} n_i(t, \mathbf{\Phi})^2 - \left( \int \mathrm{d}^2\mathbf{\Phi} \frac{|\chi_\nu(\mathbf{\Phi})|^2}{W_\nu} n_i(t, \mathbf{\Phi}) \right)^2, \tag{B.23}$$

of the classical trajectories and by two other terms:

$$[(\Delta \hat{n}_i)(t)]^2 = [(\Delta \hat{n}_i)(t=0)]^2 + \mathrm{Var}_{|\chi_\nu|^2}[n_i(t, \mathbf{\Phi})]$$

$$+ \int \mathrm{d}\mathbf{\Phi} \frac{|\chi_\nu(\mathbf{\Phi})|^2}{W_\nu} \left( g_{\nu,ii}(\mathbf{\Phi} - \omega t) - g_{\nu,ii}(\mathbf{\Phi}) \right) + \delta C_i(t), \tag{B.24}$$

where $\delta C_i(t)$ is a bounded term taking the form of correlations between the classical trajectories $n_i(t, \mathbf{\Phi})$ and a current density of the initial state $J_{\nu,i}(\mathbf{\Phi})$

$$\delta C_i(t) = 2 \int \mathrm{d}^2\mathbf{\Phi} \frac{J_{\nu,i}(\mathbf{\Phi})}{W_\nu} n_i(t, \mathbf{\Phi}) - 2 \langle \hat{n}_i \rangle_{\Psi_\nu(t=0)} \int \mathrm{d}^2\mathbf{\Phi} \frac{|\chi_\nu(\mathbf{\Phi})|^2}{W_\nu} n_i(t, \mathbf{\Phi}). \tag{B.25}$$

Let us comment the result (B.24). As discussed above, the classical trajectories $n_i(t, \mathbf{\Phi})$ characterize the spreading of the projected observables $\hat{P}_\nu \hat{n}_i \hat{P}_\nu$, and the quantum metric relates the spreading of the projected and non-projected observables. Concerning Bloch oscillations and Bloch breathing, the important feature of this quantum metric contribution is that it is small compared to the initial value $[(\Delta \hat{n}_i)(t=0)]^2$ in the case of small $\Delta\phi$, and it is vanishingly small at quasi-periods $T$. As discussed in appendix B.4, the quasi-periods are defined such that $\mathbf{\Phi} - \omega T \simeq \mathbf{\Phi}$. As a result, the quantum metric contribution vanishes at a quasi-period. The last term $\delta C_i(t)$ has the same features. It is vanishingly small at quasi-periods $T$ since $n_i(T, \mathbf{\Phi}) \simeq 0$. It is also small in the small $\Delta\phi$ limit since it can be written as classical correlations with respect to the density $|\chi_\nu(\mathbf{\Phi})|^2$ between the function $\partial_i \alpha(\mathbf{\Phi}) + A_{\nu,i}(\mathbf{\Phi})$ and $n_i(t, \mathbf{\Phi})$, with $\alpha(\mathbf{\Phi})$ the complex argument of $\chi_\nu(\mathbf{\Phi})/W_\nu$.

We thus recover the behaviors of Bloch oscillations and Bloch breathing discuss in Sec. 4.2: the spreading of the cat component remains almost constant $(\Delta \hat{n}_i)(t) \simeq (\Delta \hat{n}_i)(t=0)$ in the case of localization in phase $\Delta\phi \ll 1$, and it refocuses at quasi-period $T$, $(\Delta \hat{n}_i)(T) \simeq (\Delta \hat{n}_i)(t=0)$ irrespective to the value of $\Delta\phi$.

### B.4 Quasi-periods

We introduce the quasi-periods $T$ at which the adiabatic components refocus (for example $T = t_a, t_c$ on Fig. 6(e)) Historically, Bloch oscillations were first considered in one dimension [73]. The electric field induces a constant increase of the Bloch momenta of a semiclassical wavepacket, which crosses periodically the one dimensional Brillouin zone. As a consequence the average position of the wavepacket oscillates periodically. In two dimensions, Bloch oscillations are richer. The Bloch momenta evolves on the two-dimensional Brillouin zone along the direction of the electric field: $\mathbf{\Phi}(t) = \mathbf{\Phi}^0 - \omega t$. Such an evolution is periodic for a commensurate ratio between $\omega_1$ and $\omega_2$, corresponding to an electric field in a crystalline direction. Noting $\omega_1/\omega_2 = p_1/p_2$ with $p_1$ and $p_2$ coprime integers, the trajectory of the Bloch momenta on the two-dimensional Brillouin zone is periodic with period $T = p_1 2\pi/\omega_1 = p_2 2\pi/\omega_2$. In practice any real number $\omega_1/\omega_2$ can be approximated by a set of rational numbers [25, 75]. Each rational approximation leads to a quasi period $T$ for which $\mathbf{\Phi}^0 - \omega T \simeq \mathbf{\Phi}^0$. The times $t_a$ and $t_c$ on Fig. 6 are two examples of these quasi-periods for our choice of $\omega_1, \omega_2$.

The periodicity of a trajectory on the Brillouin zone translates into a periodic motion in the direction of the electric field $n_E$ but not in the transverse direction $n_\perp$, even in the absence of an anomalous velocity [76, 88]. For an initial phase $\mathbf{\Phi}^0$, the classical equations of motion in adiabatic space $\nu$ written in rotated coordinates reads

$$n_E(t, \mathbf{\Phi}^0) = \int_0^t \frac{1}{\hbar} \frac{\partial E_\nu}{\partial \phi_E}(\mathbf{\Phi}(t')) \mathrm{d}t', \tag{B.26}$$

$$n_\perp(t, \mathbf{\Phi}^0) = \int_0^t \left( \frac{1}{\hbar} \frac{\partial E_\nu}{\partial \phi_\perp}(\mathbf{\Phi}(t')) - |\omega| F_\nu(\mathbf{\Phi}(t')) \right) \mathrm{d}t', \tag{B.27}$$

where the evolution of the phase reads in the rotated phase coordinates: $\phi_E(t) = \phi_E^0 - |\omega| t$ and $\phi_\perp(t) = \phi_\perp^0$. The time integral can be rewritten as a line integral over $\phi_E$, leading to the conservation equation $\hbar |\omega| n_E(t, \mathbf{\Phi}^0) = E_\nu(\mathbf{\Phi}^0) - E_\nu(\mathbf{\Phi}(t))$, which is vanishingly small at a quasi-period $T$ such that $\mathbf{\Phi}(T) \simeq \mathbf{\Phi}^0$. A quasi-period defines an almost closed trajectory on the torus. The line integral of (B.27) does not vanish on this closed trajectory, and is set by the topological drift $n_\perp(T, \mathbf{\Phi}^0) \simeq -|\omega| \mathcal{C}_\nu T/(2\pi)$. The approximation gets better for longer quasi-periods $T$, *i.e.* large $p_1$ and $p_2$.

## C  Numerical method

For the numerical simulation, we diagonalize the Hamiltonian in $(n_1, n_2)$ representation, where $e^{i\hat{\phi}_i} |n_i\rangle = |n_i - 1\rangle$, with the truncation $-59 \leq n_1 \leq 59$ and $-52 \leq n_2 \leq 52$. We keep only the positions $(n_1, n_2)$ in an rectangle oriented along the directions $n_\perp$ and $n_E$ (8), corresponding to

$$|n_E| = \frac{1}{\sqrt{\omega_1^2 + \omega_2^2}} |\omega_1 n_1 + \omega_2 n_2| \leq 30, \tag{C.1}$$

$$|n_\perp| = \frac{1}{\sqrt{\omega_1^2 + \omega_2^2}} |-\omega_2 n_1 + \omega_1 n_2| \leq 50, \tag{C.2}$$

with $\omega_2/\omega_1 = (1 + \sqrt{5})/2$. We use open boundary conditions.

We construct numerically the adiabatic projector up to order 1 in the adiabatic parameter $\epsilon$. The adiabatic projector $\hat{P}_\nu$ is defined by an asymptotic series in the formal dimensionless parameter $\epsilon$

$$\hat{P}_\nu = \sum_{r=0}^{\infty} \epsilon^r \hat{P}_{\nu,r} = \hat{P}_{\nu,0} + \epsilon \hat{P}_{\nu,1} + \dots, \tag{C.3}$$

such that

$$[\hat{H}_{\text{tot}}, \hat{P}_\nu] = 0, \tag{C.4}$$

$$\hat{P}_\nu \hat{P}_\nu = \hat{P}_\nu, \tag{C.5}$$

with

$$\hat{H}_{\text{tot}} = H(\hat{\phi}_1, \hat{\phi}_2) + \epsilon(\omega_1 \hat{n}_1 + \omega_2 \hat{n}_2). \tag{C.6}$$

We use the half-BHZ model for the qubit (5) with the gap parameter $\Delta = 2$. The maximum on $(\phi_1, \phi_2)$ of the ground state energy of $H(\phi_1, \phi_2)$ for these values of parameters is $E^0_{-,\text{max}} = -1$, and the minimum of excited state energy is $E^0_{+,\text{min}} = 1$.

At order 0, $\hat{P}_\nu$ is a spectral projector of the Hamiltonian $H(\hat{\phi}_1, \hat{\phi}_2)$. We diagonalize numerically the Hamiltonian at zero frequency $H(\hat{\phi}_1, \hat{\phi}_2)$ in the truncated Hilbert space

$$H(\hat{\phi}_1, \hat{\phi}_2) \big| \Psi^0_k \big\rangle = E^0_k \big| \Psi^0_k \big\rangle. \tag{C.7}$$

The projector at order 0 is the projector on the states of the ground band, i.e. on the states such that $E_k < E^0_{-,\text{max}}$

$$\hat{P}_{-,0} = \sum_{\substack{k \\ E_k < E^0_{-,\text{max}}}} \big| \Psi^0_k \big\rangle \big\langle \Psi^0_k \big|. \tag{C.8}$$

Note that we do not take into account the edge states whose energy lie in the gap for the construction of the projectors.

The conditions (C.4) and (C.5) translates in recursive conditions for the different orders $\hat{P}_{\nu,r}$ of the projector

$$\hat{P}_{\nu,r} = \sum_{s=0}^{r} \hat{P}_{\nu,s} \hat{P}_{\nu,r-s}, \tag{C.9}$$

$$[H(\hat{\boldsymbol{\phi}}), \hat{P}_{\nu,r}] = [\hat{P}_{\nu,r-1}, \hbar\omega \cdot \hat{\mathbf{N}}], \tag{C.10}$$

from which we can deduce order by order the expression of $\hat{P}_{\nu,r}$ in the basis of $\big| \Psi^0_k \big\rangle$. At order 1 we obtain

$$\hat{P}_{-,1} = \sum_{\substack{k,l \\ E_k < E^0_{-,\text{max}} \\ E_l > E^0_{+,\text{min}}}} \big| \Psi^0_k \big\rangle \frac{\big\langle \Psi^0_k \big| \hbar\omega \cdot \hat{\mathbf{N}} \big| \Psi^0_l \big\rangle}{E^0_k - E^0_l} \big\langle \Psi^0_l \big| + \text{h.c.}, \tag{C.11}$$

which can be constructed numerically.

# D  Difference between eigenstates and adiabatic states

We show that after the time of separation $t_{\text{sep}}$, the cat component which splits in the direction $n_\perp < n^0_\perp$ identifies with the adiabatic component $\hat{P}_- |\Psi(t)\rangle$ of the total state. The adiabatic projector $\hat{P}_-$ is constructed from the qubit's adiabatic states $|\psi_-(\boldsymbol{\Phi})\rangle$ (9). It is a perturbative correction of the spectral projector of the qubit Hamiltonian $H(\hat{\boldsymbol{\Phi}})$ constructed from the eigenstates $\big| \psi^0_-(\boldsymbol{\Phi}) \big\rangle$. This enables us to quantify the difference between eigenstates and adiabatic states.

We note $\hat{P}_<$ the projector on the states $|n_1\rangle \otimes |n_2\rangle \otimes |s\rangle$ with $n_\perp = (-\omega_2 n_1 + \omega_1 n_2)/|\omega| < n^0_\perp$, $s = \uparrow_z, \downarrow_z$, and $|\Psi_<(t)\rangle = \hat{P}_< |\Psi(t)\rangle$ the component of the system on this region $n_\perp < n^0_\perp$.

The adiabatic projector $\hat{P}_-$ is constructed numerically at order 0 $\hat{P}_-^{(0)} = \hat{P}_{-,0}$ and at order 1 $\hat{P}_-^{(1)} = \hat{P}_{-,0} + \hat{P}_{-,1}$ from the expressions (C.8) and (C.11). We note respectively $\left|\Psi_-^0(t)\right\rangle = \hat{P}_-^{(0)}\left|\Psi(t)\right\rangle$ and $\left|\Psi_-^1(t)\right\rangle = \hat{P}_-^{(1)}\left|\Psi(t)\right\rangle$ the adiabatic projections respectively at order 0 and order 1 of the total state $|\Psi(t)\rangle$.

We note $F(\Psi_1, \Psi_2) = \langle\Psi_1|\Psi_2\rangle/(\langle\Psi_1|\Psi_1\rangle\langle\Psi_2|\Psi_2\rangle)$ the fidelity between two states $|\Psi_1\rangle$ and $|\Psi_2\rangle$. The figure 7(a) represents the fidelity $F(\Psi_<, \Psi_-^0)$ and $F(\Psi_<, \Psi_-^1)$ between the cat component $|\Psi_<(t)\rangle$ and the adiabatic projections at order 0 and order 1. After the time of separation $t_{\text{sep}} \simeq 8$, the cat component has a fidelity of approximately 99.5% with $\left|\Psi_-^0(t)\right\rangle$ and 99.9% with $\left|\Psi_-^1(t)\right\rangle$. The adiabatic projection at order 0 gives a very good approximation of the cat component, corrected at higher orders to gives the full adiabatic component $|\Psi_-(t)\rangle$. The slight decrease with time of the fidelity after the time of separation is due to the successive Landau-Zener transitions.

The difference between the adiabatic projector $\hat{P}_-$ and the spectral projector $\hat{P}_-^{(0)}$ is also visible in the weight of the cat state. We represent on Fig. 7(b) the weight of the adiabatic projection computed dynamically from the splitting $W_- = \left\langle\Psi_<(t_{\text{sep}})\middle|\Psi_<(t_{\text{sep}})\right\rangle$, and the weight $W_-^0 = \left\langle\Psi_-^0\middle|\Psi_-^0\right\rangle$ and $W_-^1 = \left\langle\Psi_-^1\middle|\Psi_-^1\right\rangle$ computed numerically from the projection of the initial state respectively at order 0 and order 1. The initial state is a Gaussian state centered on $\Phi^0 = (0,0)$ and the qubit in $(|\uparrow\rangle + |\downarrow\rangle)/\sqrt{2}$. The first order correction $W_-^1$ almost identifies with the weight obtained dynamically $W_-$.

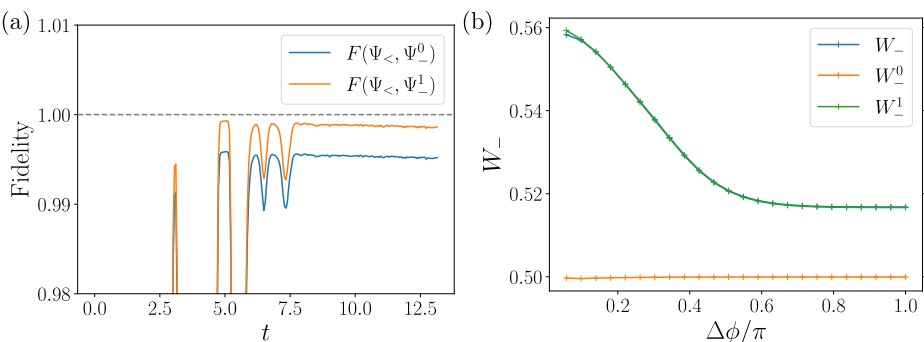

Figure 7: Difference between ground projection and adiabatic projection. (a) Fidelity $F(\Psi_<, \Psi_-^0)$ and $F(\Psi_<, \Psi_-^1)$ between the cat component $|\Psi_<(t)\rangle$ and the adiabatic projections at order 0 and order 1 $|\Psi_-^{0/1}(t)\rangle$. After the time of separation $t_{\text{sep}} \simeq 8$, the cat component $|\Psi_<(t)\rangle$ is very close to the lowest order adiabatic approximation $\left|\Psi_-^0(t)\right\rangle$ with a fidelity of 99.5%. The fidelity further increases with the adiabatic projection of order 1, such that $|\Psi_<(t)\rangle$ identifies with $|\Psi_-(t)\rangle$. (b) Effect of the difference between eigenstates and adiabatic states on the weight of the cat. Initial state of the Fig. 4(c) with the qubit prepared in $(|\uparrow\rangle + |\downarrow\rangle)/\sqrt{2}$. In blue, weight $W_-$ of the cat computed dynamically from the splitting (same as Fig. 4(c)). In orange, weight $W_-^0 = \left\langle\Psi_-^0\middle|\Psi_-^0\right\rangle$ computed from the spectral projector $\hat{P}_-^{(0)}$. In green, weight $W_-^1 = \left\langle\Psi_-^1\middle|\Psi_-^1\right\rangle$ from the projector at order 1, which almost identifies with the weight obtained dynamically.

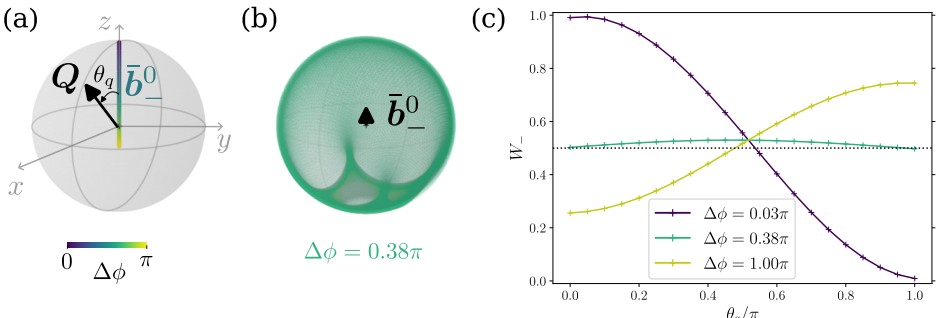

Figure 8: Two types of preparation of the qubit and the modes inducing a symmetric adiabatic cat states $W_\pm = 1/2$. (a) **Q** the qubit initial state parametrized by its longitudinal angle $\theta_q$. $\Delta\phi$ the width in phase of the initial Gaussian state, and $\bar{\mathbf{b}}^0_-$ the corresponding average ground state (perturbatively close to the average adiabatic state $\bar{\mathbf{b}}_-$ of Fig. 4). (b) For $\Delta\phi = 0.38\pi$ the density adiabatic states cover almost uniformly the Bloch sphere (in green), inducing an almost vanishing average adiabatic state $\bar{\mathbf{b}}_- \simeq 0$. (c) First type of symmetric cat state obtained by preparing the qubit **Q** orthogonally to $\bar{\mathbf{b}}_-$ ($\theta_q \simeq \pi/2$). Second type of symmetric cat state when $\bar{\mathbf{b}}_- = 0$, obtained with $\Delta\phi \simeq 0.38\pi$ irrespective of the qubit state **Q**.

# E  Symmetric cat states

## E.1  Two types of symmetric cat states

Following the analysis of Sec. 3.3, we can identify two types of separable initial states that give rise to symmetric cat states with $W_+ = W_- = \frac{1}{2}$. The first class is obtained by preparing the qubit orthogonally to the average adiabatic state $\bar{\mathbf{b}}_\pm$. For our initial phase $\mathbf{\Phi}^0$, the average of eigenstates $\bar{\mathbf{b}}^0_- = \int d^2\mathbf{\Phi} \, |\chi(\mathbf{\Phi})|^2 \mathbf{b}^0_-(\mathbf{\Phi})$ lies on the $z$-axis for all $\Delta\phi$ by symmetry of the model (5) (see Fig. 8(a)). The adiabatic state are perturbatively close to the eigenstates, such that $\bar{\mathbf{b}}_\pm$ is perturbatively closed to the $z$-axis. Hence a qubit prepared on the equator of the Bloch sphere ($\theta_q = \pi/2$ on Fig. 8(a)) corresponds to two almost equal weights for all $\Delta\phi$, see Fig. 8(c). The deviation from $W_- = 1/2$ at $\theta_q = \pi/2$ originates from the difference between the eigenstates and the adiabatic states detailed in appendix D.

The second class of symmetric cat states is obtained for well-chosen Gaussian states of the modes, for which $\bar{\mathbf{b}}_\pm = 0$. In such case, $W_\pm = 1/2$ for any initial state of the qubit. Let us illustrate this case. For $\Delta\phi = 0.38\pi$, the density adiabatic states cover almost uniformly the Bloch sphere (in green on Fig. 8), inducing an almost vanishing average adiabatic state $\bar{\mathbf{b}}_- \simeq 0$. Thus, the cat has (almost equal) weights $W_+ = W_-$ independently of the initial state of the qubit $\theta_q$ (Fig. 8(c) green curve).

## E.2  Purity from initial Fock state

We illustrate the role of the phase densities in the entanglement between the qubit and the modes in the cat components $|\Psi_\pm(t)\rangle$ for a quasi-Fock initial state $\Delta\phi = \pi$. The qubit is initialized in $\theta_q = \pi/2$, inducing a symmetric cat as discussed in the previous section.

Given (20) the reduced density matrix of the qubit corresponds to an average of adiabatic states with respect to translated phase densities $|\chi_\pm(\mathbf{\Phi} + \omega t)|^2$. From (11), the densities of the two components satisfy

$$|\chi_+(\mathbf{\Phi} + \omega t)|^2 + |\chi_-(\mathbf{\Phi} + \omega t)|^2 = |\chi(\mathbf{\Phi} + \omega t)|^2. \tag{E.1}$$

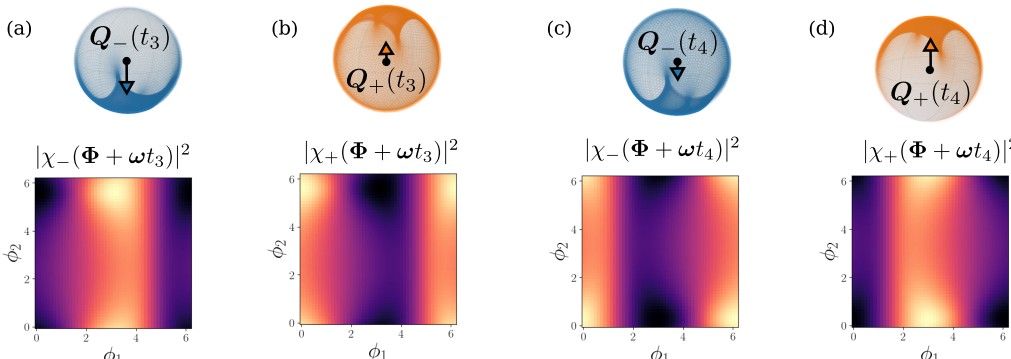

Figure 9: Phase densities $|\chi_\pm(\boldsymbol{\Phi} + \omega t)|^2$ of the cat components $|\Psi_\pm(t)\rangle$ translated to densities of adiabatic states on the Bloch sphere via the map $\boldsymbol{\Phi} \mapsto \mathbf{b}_\pm(\boldsymbol{\Phi})$. The polarization of the qubit $\mathbf{Q}_\pm(t)$ is the average of adiabatic states with respect to these densities.

Hence they split the phase density of the total system $|\chi(\boldsymbol{\Phi} + \omega t)|^2$ in two complementary supports.

We represent these densities on Fig. 9 at $t = t_3$ and $t = t_4$ (see Fig. 5(b1)). The densities on the torus translate into densities of adiabatic states on the Bloch sphere via the map $\boldsymbol{\Phi} \mapsto \mathbf{b}_\pm(\boldsymbol{\Phi})$. At $t = t_3$, $|\chi_-(\boldsymbol{\Phi} + \omega t_3)|^2$ covers a smaller part of the Bloch sphere (in blue on Fig. 9(a)) than $|\chi_+(\boldsymbol{\Phi} + \omega t_3)|^2$ (in orange on Fig. 9(b)). As a result, the qubit is more entangled with the modes in $|\Psi_+(t_3)\rangle$ than in $|\Psi_-(t_3)\rangle$: $|\mathbf{Q}_+(t_3)| < |\mathbf{Q}_-(t_3)|$. During the dynamics, the phase densities are translated on the torus with no dispersion, changing the densities on the Bloch sphere and the purity of the qubit. At $t = t_4$, the domains have been almost exchanged $|\chi_-(\boldsymbol{\Phi} + \omega t_4)|^2 \simeq |\chi_+(\boldsymbol{\Phi} + \omega t_3)|^2$, such that $\mathbf{Q}_-(t_4) \simeq -\mathbf{Q}_+(t_3)$ and $\mathbf{Q}_+(t_4) \simeq -\mathbf{Q}_-(t_3)$.

For all the figures of the manuscript, the densities are computed from the eigenstates $\left|\psi_-^0(\boldsymbol{\Phi})\right\rangle$, providing a qualitatively accurate illustration of the densities of adiabatic states $|\psi_-(\boldsymbol{\Phi})\rangle$.

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
