# Peer review of "Topological dynamics of adiabatic cat states"

_SciPost Physics, doi:SciPost Phys. 17, 112 (2024)_

## Round 1 · Referee Report · Anonymous (Referee 1) · 2023-3-31

Strengths

  • Thorough analytic and numerical study

Weaknesses

  • Phenomenon described in the paper already discovered

Report

The manuscript “Topological dynamics of adiabatic cat states” studies the generation of cat states in a 2-level system (a spin) coupled to 2 driving modes. The cat states are generated using the mechanism of topological frequency conversion from Ref 14, where an adiabatically 2-tone driven spin acts as a medium for energy transfer between the tones at a universal rate; the direction of energy transfer is determined by the alignment of the spin with the instantaneous net field acting on the modes. When aligning the spin perpendicular to the initial field (Ie in an equal superposition of alignment and anti-alignment), the “photon numbers” of the individual modes change in time at opposite rates for the aligned and anti-aligned components. This paper studies how this principle can be used to generate a cat state which is in an equal superposition of two states with highly distinct photon numbers.
The idea put forward in the paper, i.e., of using topological frequency conversion to generate cat states, has been proposed and studied previously in the literature, namely in Refs. [22] (and also in [24]). It is therefore not clear what constitutes the new discovery in the present manuscript. The present paper does provide a thorough analytic/numerical characterization of the cat states; however, I currently do not see a clear motivation for this analysis that would justify publication based on this characterization alone.
Therefore, I unfortunately do not see how the manuscript meets any of SciPost’s 4 expectations (listed under acceptance criteria). I would reconsider the manuscript if (1) the authors clearly explain what constitutes their new discovery, in particular how it contrasts to, e.g., Refs. 22 and 24., and (2) how this discovery meets the acceptance criteria of Scipost.
As a more physics-related question, I am a little unsure of how ``cattiness’’ manifests itself physically. As I understand, the operators n_1 and n_2 (which distinguish the two halves of the cat state) do not directly correspond to physical observables, but are mathematical constructs introduced to conveniently describe multi-tone driven quantum systems. The authors do mention that they for instance emerge when treating the drives as quantized modes and working in the large-n limit. However, I think the manuscript would benefit greatly if the authors made the physical meaning of these observables more clear, and, even better, proposed ways to detect the cattiness.
  • validity: ok
  • significance: poor
  • originality: poor
  • clarity: ok
  • formatting: good
  • grammar: below threshold

Author:  Jacquelin Luneau  on 2024-05-07  [id 4478]

(in reply to Report 1 on 2023-03-31)

Please see the attached response.

Attachment:

referee1.pdf

Author:  Jacquelin Luneau  on 2023-04-04  [id 3543]

(in reply to Report 1 on 2023-03-31)
Category:
reply to objection

We would like to provide clarifications regarding the originality of our work. On the technical side, our work is the first one to treat on equal footing the driving modes and the qubit by using a full quantum description of both modes, contrary to [22] and [24], which rely on Floquet theory and therefore treat one drive as a classical parameter of the Hamiltonian. Thus, on the technical side the techniques employed are different. In particular, we develop an adiabatic approximation valid to all orders in the adiabatic perturbative parameter, which is original.

On the physical side, our full quantum treatment enables a quantitative description of (i) the entanglement between the qubit and the two quantum modes which was not provided before, (ii) the role of the quantum metric and of topological constraint on the nature of the adiabatic states, (iii) the time evolution of the quantum fluctuations of the modes' photon numbers. None of these important properties of the dynamics of the model were discussed previously.

Beside, let us stress that we denote cat state a superposition of two states distinguishable through measures of the modes’ energy, but not necessarily an equal weight superposition.
In this sense we show that any realistic separable non-cat initial state dynamically evolves into a cat state. In contrary [22] says only that some well-prepared initial state evolves into a cat state and [24] does not discuss the creation of a cat state by the dynamics but the time evolution of an initial cat state, which is a very different phenomenon. In short, we show that cat states are generic and not accidental. The understanding of the generic nature of the creation of cat states results from our precise description of the adiabatic projector, which is possible only within a fully quantum mechanical treatment of both cavity modes, such as ours.

---

## Round 1 · Referee Report · Anonymous (Referee 2) · 2023-5-16

Strengths

The topic is timely. The conclusions are based on thorough derivations and detailed arguments.

Weaknesses

The paper ends up very heavy to read since it focuses much on technical details. For the final results -- utilizing a quantized version of a topological pumping one can prepare cat states -- it seems unmotivated to fill 33 pages consisting much of derivations (cat states are pretty general in physical models). The specific model considered seems unnecessarily complicated if the goal is to prepare cat states.

Report

In the manuscript "Topological dynamics of adiabatic cat states", Luneau et al. consider a qubit coupled to two boson modes. In particular, the setup with two quantized modes driving the qubit generalizes a model of topological pumping with two classical fields. Using analytical arguments and derivations, accompanied by numerical simulations, it is shown how the evolution results in cat states characterized by a superposition of states well separated in phase space.

The paper provides a thorough analysis of the system at hand, their arguments are exhaustive and detailed. The objective of the paper, analyzing how the topological pumping mechanism is modified by replacing the classical fields with quantized ones, feels relevant. The conclusions are backed up by a solid analysis. Nonetheless, personally, I find this paper very hard to read, not due to the complexity of the problem, but because I drown in technical details. While refereeing I had to split the reading between several occasions in order to appreciate all arguments. Often, while reading all the technical analyzes I had to return to the beginning of the paper to remind myself what was actually the purpose of the paper; instead of following a logic line with clear goals I got stuck in derivations (one minor reason for this was that some notations were not what I am used to, but that varies from communities and personal taste). I had the feeling that many of the technical derivations could be included as appendices, but then I noticed that there were already nine appendices. I understand that this is a very personal opinion, and some others might appreciate all the technical parts, but my feeling is that to get the message through one does not need all this. After all, the main conclusion that a qubit interacting with two quantized modes results in cat states is not very new, but rather general. In fact, two boson modes initially in coherent states and interacting with a qubit via a Jaynes-Cummings interaction will also result in an entangled cat. Apart from the above general comment about the structure I have some more specific comments:

1) I lack a discussion about the relevance of the results. The light-matter interaction defined in eq. (2) involves the phase operators, and seems very particular. The meaning of phase states has a long history in quantum optics; both in terms of how to define and interpret a conjugate variable to the number operator, and how phase states can be prepared. Typically, the phase operators when expressed in terms of Fock states involve infinitely large Fock states, and I do not know how realistic the interaction of eq (2) is for real experiments. How could it be realized? As said above, it seems that cat states can be prepared by much simpler means.

2) The authors stress how the topological pumping with quantized fields leads to the build-up of entanglement between the qubit and the fields. Of course, this is a general statement and does not depend on topology. Indeed, others have asked similar questions in the past; what if we replace the classical field with a quantum one? The authors cite a few works already, and another one is Rev. Lett. 89, 220404 (2002).

3) The link to Bloch oscillations is mentioned. It brings to mind a recent work of Bloch oscillations in state space Rev. A 98, 053820 (2018).

Requested changes

Some specific suggestions are mentioned in my report.

  • validity: good
  • significance: good
  • originality: ok
  • clarity: low
  • formatting: reasonable
  • grammar: good

Author:  Jacquelin Luneau  on 2024-05-07  [id 4479]

(in reply to Report 2 on 2023-05-16)

Please see the attached response.

Attachment:

referee2.pdf

---

## Round 2 · Referee Report · Anonymous (Referee 2) · 2024-8-16

Strengths

1) Thorough detailed analysis. 2) Timely topic.

Weaknesses

1) The main results hidden behind technical derivations/discussions. 2) Its relevance not well explored. 3) Somewhat lack of novelty.

Report

In my initial report, my primary concern was that the paper's main results were obscured by lengthy technical derivations. I am pleased to see that this issue has been largely resolved in the revised version. The revised manuscript now places a greater emphasis on the conceptual issues, rather than being weighed down by excessive detail. Although I still believe the paper could benefit from further streamlining by omitting some of the less central results to focus more sharply on the key findings, I recognize that this is a subjective opinion. I acknowledge that some readers may appreciate the detailed, comprehensive approach.

The manuscript is well-written, and the topic is of broad interest. On this basis, I am inclined to recommend the paper for publication. However, while reviewing the revised manuscript, a few additional points came to mind that I would like the authors to consider before finalizing the paper:

1) The authors discuss the significance of exact or approximate phase states, and I have the following thought. Rather than using canonical phase and amplitude variables, we could instead consider quadrature operators x and p. This substitution would modify the Bloch Hamiltonian in equations (5) to (up to commutation relations):

hx=Δx/sqrt(nx), hy=Δy/sqrt(ny), hz=Δ(2−px/sqrt(nx)−py/sqrt(ny))/4.

By ignoring fluctuations in nx​ and ny​, this simplifies the model significantly, making it easier to realize experimentally. Moreover, a similar model, with hz=Δ/2, has been previously studied and it directly follows how it can be used for generating cat states (see Phys. Rev. A 81, 051803(R) (2010)). This simplified model has also been recently realized experimentally (see Nature Chemistry 15, 1509 (2023) and Nature Chemistry 15, 1503 (2023)). My question is: To what extent do the present results hold in these simpler models? For example, are the cos⁡(ϕ1) and cos⁡(ϕ2) terms in hz essential?

2) Considering the previous question, it seems that the intriguing physics of the model described in equations (5) arises from a conical intersection, which leads to non-trivial topology. Conical intersections are a well-established topic in molecular and chemical physics, with their associated gauge structures extensively studied. I wonder whether the analysis presented in this paper could be linked to this broader area of physics.

I think that addressing these two points could further enhance the impact and scope of the manuscript.

Requested changes

I would appreciate it if the authors could address the two questions I raised in my report. If these questions prove to be irrelevant, they don't necessarily need to be included in the manuscript.

Recommendation

Ask for minor revision

  • validity: ok
  • significance: ok
  • originality: ok
  • clarity: ok
  • formatting: reasonable
  • grammar: good

Author:  Jacquelin Luneau  on 2024-09-13  [id 4778]

(in reply to Report 1 on 2024-08-16)

Please see the attached response.

Attachment:

referee1.pdf

---

## Round 2 · Referee Report · Anonymous (Referee 1) · 2024-8-20

Strengths

1 - method and results are valid 2- relation to quantum geometry and elucidation of role of entanglement is interesting

Weaknesses

1- Paper is very long, and its length and the extensive amount of details covered make it hard to appreciate the results, which are interesting, but could be presented in a clearer and more concise fashion. 2- The "fully quantum" treatment of the modes in the present manuscript appears to be equivalent to a treatment where the modes are fully classical, since the large-n limit is taken.

Report

The manuscript improved considerably, and the authors have addressed most of my concerns from the last report. I believe the manuscript can be published provided the following points are addressed.

  1. The authors state that their adiabatic expansion scheme is valid to all orders in the modes' frequencies. However, in the Appendix where the expansion is provided, it is noted that the expansion is asymptotic (i.e., that it will diverge beyond a certain order for any finite parameter). To some, the statement that the expansion is "valid to all orders" could imply that the expansion converges for any choice of input parameter. I therefore suggest that the authors clarify in the main text that the expansion is asymptotic.

  2. The rotor model considered in Eq. (4) is equivalent to the classical drive model, since the large-photon-number limit is taken (i.e., the Hamiltonian in Eq. (4) is equivalent to the Sambe (or extended Hilbert) space representation of the multi-mode classical drive problem, e.g. considerd in Ref. 15. This seems to contradict the authors claim that their treatment is "fully quantum" in contrast to earlier works. I suggest the authors modify their wording to clarify that the model they consider is equivalent to a classical-drive model, and has also been studied in earlier works. The difference to earlier work is that they consider the quantum correlations of the classical drives [in the sense that a classical model is recovered from a quantum model by taking the hbar->0 limit, even while the Schrodinger equation still describes the system in this limit].

Recommendation

Ask for minor revision

  • validity: good
  • significance: ok
  • originality: low
  • clarity: low
  • formatting: reasonable
  • grammar: good

Author:  Jacquelin Luneau  on 2024-09-13  [id 4777]

(in reply to Report 2 on 2024-08-20)

Please see the attached response.

Attachment:

referee2.pdf

---

## Round 2 · Author Response

Dear editor,

We would like to thank you for your valuable comments and suggestions concerning the improvements on our manuscript that we have decided to resubmit to SciPost Physics.
We followed your suggestion of a major revision of our manuscript to clarify the main points and
precise their relation to earlier work.

The first outcome of our work is the notion of topological coupling between quantum degrees of freedom, a qubit and two bosonic modes. Our work is the first one to treat within a quantum description both driving modes and the qubit, contrary to Refs. 23,25 which rely on Floquet theory and therefore treat one drive as a classical parameter of the Hamiltonian.
Our full quantum treatment enables a quantitative description of (i) the entanglement between the qubit and the two quantum modes, (ii) the role of the quantum metric and of topological constraint on the nature of the adiabatic states, (iii) the time evolution of the quantum fluctuations of the modes' photon numbers and its relation with Bloch oscillations and Bloch breathing. None of these important properties of the dynamics of the model were discussed previously.
On the technical side, we develop an adiabatic expansion valid to all orders in the adiabatic perturbative parameter, which is original. This reveals the role of a "dressed" Berry curvature in the dynamics instead of the bare Berry curvature. This also enables us to show that (iv) the adiabatic states are not naturally prepared experimentally but generically evolve into
cat states.

We believe that our discoveries meet the acceptance criteria 2 of SciPost Physics. Indeed, the starting point of our work consists in opening a new pathway in the study of topological pumping by considering quantized modes coupled to a qubit instead of a classically driven qubit. In doing so we rely on a separation between fast and slow quantum degrees of freedom instead of a Floquet formalism.
Our change of perspective enables us to demonstrate several fundamental properties of the dynamics of this coupled system. In particular, we unveil topological constraints on the entanglement between the qubit and the modes. This opens a new pathway on the relation between topology and entanglement.

Besides, in the context of Thouless pumps, this opens a new pathway in the understanding of the breakdown of quantization of the pumped charge due to the splitting in a cat state.
For instance one can build on our results to propose protocols of preparation of the adiabatic states which leads to a quantized pumping. We now present these perspectives in the conclusion.

We thank you for your kind consideration.
With our respectful regards,
Jacquelin Luneau,
Benoît Douçot,
and David Carpentier.

---

## Round 2 · List of Changes

Every section has been significantly modified to highlight their main original results. We changed the section 2 to answer valuable questions of the referees concerning how our model of quantum rotor can describe quantum optical system, and what is the physical interpretation of the operators n1 and n2. To clarify these aspects, we added the section 2.1.1 in which we define the notion of topological coupling between two bosonic modes and a two-level system. Doing so, we consider usual types of quantum-optical coupling between a two-level system and the quadratures bosonic modes. The operators n1 and n2 are then naturally the number of quanta in each harmonic oscillator. We added the figure 1 to illustrate this model and the regime of topological coupling, which has never been considered before. We then explain in details in section 2.1.2 that our model of quantum rotors accurately describes the dynamics of quantum states in which the average number of quanta is large compared to their quantum fluctuations. Furthermore, we added a paragraph at the end of Sec. 2.1 establishing the dictionary between our model and a Rice-Mele model with quantum drive. Thus, our results also describe the effect of the quantum nature of a drive on a Thouless pump, which, to our knowledge, has not been discussed before. We added paragraphs translating our results in this context of quantum Thouless pumps at the end of each main section of the manuscript.

In Sec. 3.1, we moved most the technical details in appendix A to highlight the originality of our approach concerning the adiabatic theory: we focus on the nature of the states governed by the effective adiabatic equations of motion, called the adiabatic states. This enables us to show that they are not naturally prepared experimentally, but generically any initial state decomposes into a pair of those, leading to a splitting into a cat state at a topologically quantized rate. We simplified Fig. 4 (former Fig. 3) and Sec. 3.2 on the weight of the cat components to stress the main point: for small quantum fluctuations of the driving phase, this weight is governed by the quantum metric, and the weight of both components are comparable for an initial Fock state due to the topological nature of the coupling. In particular, we moved the secondary results of section 3.3.2 on the preparation of symmetric cat states in appendix E.

In Sec. 4.1, we removed technical derivations and removed the former Fig. 4(d,c) to stress its main results. The entanglement between the qubit and the modes is governed by the geometry of their coupling and enhanced by its topological nature. For small quantum fluctuations of the phase, the entanglement is characterized by the quantum metric. For large quantum fluctuation of the phase (quasi-Fock state), the topological nature of the coupling imposes a strong entanglement.

In Sec. 4.2, we significantly simplified the Fig. 6 (former Fig. 5) to highlight our main results on the quantum fluctuations and temporal evolution of the number of quanta. By increasing the quantum fluctuation of the phases of the mode, we go from a regime of Bloch oscillations to a regime of Bloch breathing. In particular, the quantum fluctuations of the phase stabilizes the topological pumping by reducing the amplitude of the time variation of the center of wavepacket around the topologically quantized drift. In particular, we moved the former subsidiary section 4.2.2 on quasi-periods in appendix B4.

All the modified text appears in blue in the new manuscript.

  • We rewrote the abstract. We now precise that our system is the quantum analog of topological pumping. We highlight our results on the characterization of the entanglement between the qubit and the modes by the quantum metric, and on the enhancement of entanglement by the topological nature of the coupling.

  • We added a paragraph in the introduction to stress that our definition of cat states does not require an equal superposition of two adiabatic components. We explain that the topological separation in an adiabatic cat states at quantized rate is topologically robust and occurs for any initial state, in contrast with the quantization of pumping which is known to require a well-prepared initial state.

  • We added a paragraph in the introduction to highlight our result on the role of the quantum Fubini-Study metric in the entanglement between the qubit and the modes, and to stress that a topological coupling necessarily induces a strong entanglement between a pump and its driving modes.

  • We added a paragraph in the introduction detailing the relation of our results with previous works.

  • We added the section 2.1.1 to define a notion of topological coupling between two harmonic oscillators and a qubit, in order to clarify the physical meaning of the number of quanta as suggested by Referee 1, and to recover more common types of light-matter interaction as suggested by Referee 2.

  • We added the figure 1 to illustrate the model and the regime of topological coupling between two quantum harmonic oscillators and a two-level system.

  • We added a paragraph in Sec. 2.1.2 to explain under which conditions our model of quantum rotors is an accurate description of the model of quantum harmonic oscillators, as suggested by Referee 2.

  • In Sec. 2.1.2, we merged in one paragraph the discussions of the two possible representations of the state of the modes (the phase and number representation) and the analogy with a particle on a lattice. We explain that the evolution in number representation is solely due to the coupling of the modes to the qubit.

  • We moved the numerical values of the Hamiltonian parameters from Sec. 2.2 to Sec. 2.1.2, to provide them after the introduction of the model.

  • In Sec. 2.1.2, we added a paragraph detailing the equivalence between our model and a Rice-Mele model of Thouless pump with quantum drive, to stress that our results also describe the effect of the quantum nature of a drive on a Thouless pump.

  • We removed the last paragraph of Sec. 2 detailing the organization of the remaining of the article, to avoid redundancies with the end of the introduction.

  • At the beginning of Sec. 3, we added a paragraph to stress the originality of our approach concerning the adiabatic theory: we focus on the nature of the states governed by the effective adiabatic equations of motion, called the adiabatic states. This enables us to show that they are not naturally prepared experimentally, but generically any initial state decomposes into a pair of those, leading to a splitting into a cat state at a topologically quantized rate.

  • In Sec. 3.1, we now provide only the physical meaning of the adiabatic states and moved the technical details on the definition of the adiabatic projector in appendix A.

  • In Sec. 3.1, we added a paragraph to discuss the difference between the adiabatic states and the states considered in previous hybrid classical-quantum descriptions of topological pumps, in terms of the quantum fluctuations of the phase and of the perturbative corrections of the eigenstates in the modes' frequencies.

  • In Sec. 3.1., we now stress that the curvature of the adiabatic states is a dressed Berry curvature, in the sense of a perturbative correction of the Berry curvature of the eigenstates.

  • We simplified section 3.2 to highlight its main result: any initial state evolves into a cat state splitting at a quantized rate. We moved the expression of the pumping rate in appendix B, and we kept a paragraph discussing the role of the dressed Berry curvature instead of the canonical Berry curvature in the dynamics: the dressed Berry curvature modifies the details of the temporal oscillations of the wavepacket, but does not change the average quantized drift.

  • We added a paragraph in Sec. 3.2 to discuss the relation between the topological splitting in a cat states and the breakdown of quantization of a topological pump.

  • We removed technical details on the qubit's state representation at the beginning of section 3.3 to summarize the main qualitative result this section: the weights of the two cat components are controlled by the quantum metric for small quantum fluctuations of the phase, and are comparable for large quantum fluctuations of the phases of the modes, due to the topological nature of the coupling.

  • We moved Sec 3.3.2 on symmetric cat states to appendix E.

  • We significantly simplified figure 4 (former Fig. 3) to highlight the main results of section 3.3. We moved the former Fig. 3(d) concerning symmetric cat states in Fig. 8 of appendix E.

  • We added a paragraph in the end of Sec. 3.3 to discuss the relations between our quantitative results on the weight of the cat state and previous studies on the breakdown of quantization of topological pumps.

  • We changed the title of Sec. 4 from "Characterization of cat components" to "Entanglement and dynamics of cat states".

  • We simplified section 4.1 and Fig. 5 to highlight their main result: (i) the entanglement between the modes and the qubit is controlled by the geometry of the adiabatic states (their quantum metric), and (ii) the topological nature of the coupling leads to a strong entanglement. We removed the subfigures (c) and (d) from Fig. 5 (former Fig. 4)

  • We removed technical details at the beginning of Sec. 4.1 and keep the main quantitative result Eq. (20): the qubit is in a statistical mixture of the adiabatic states weighted by the phase density of the modes.

  • We simplified section 4.1.2 on the entanglement of quasi-Fock states. We moved the details on the phase distributions of each cat component in appendix E.2 to stress the main result of this section: due to the topological nature of the coupling, large quantum fluctuations of the phase of the modes induce a high entanglement between the qubit and the modes.

  • At the beginning of Sec. 4.2, we added references on the manifestation of Bloch oscillations in artificial lattices.

  • In Sec. 4.2.1, we simplified the notation of the average and spreading of the number of quanta.

  • We moved the former section 4.2.2 on quasi-periods to appendix B.3.

  • We modified section 4.2.2 to highlight its main result: topological pumping between quantum modes is "more quantized" than topological pumping between classical modes.

  • We moved the discussion on Wannier states leading to no temporal fluctuations of pumping in a separate paragraph of Sec. 4.2.2 for clarity.

  • We simplified section 4.2.3 by moving the technical expression of the variance of classical trajectories in appendix B.3, and keeping only the discussion of its role in the refocusing of the wavepacket.

  • We changed figure 6 (former Fig. 5) on Bloch oscillations and Bloch breathing to clarify the main points. We replaced the former figures 5(a-d) by Fig. 6(a) containing the definition of the quantities studied in the figure (the average and spreading of number of quanta), and by the insets of Fig. 6(b,d) illustrating the breathing of cat components. We removed the former figures 5(e,g,j) to keep only figures 6(b-e) illustrating the main results of the section: by increasing the quantum fluctuation of the phase, we reduce the amplitude of the Bloch oscillations of the number of quanta and increase the amplitude of their Bloch breathing.

  • We moved the role of adiabatic projection on refocusing of adiabatic states to a separate paragraph of Sec. 4.2.3 for clarity.

  • In the conclusion, we now highlight our results on adiabatic perturbation theory to all orders and on the relation between entanglement and topology.

  • In the conclusion, we added a paragraph showing how our results shed light on the effects of the quantum nature of the drive of a Thouless pump.

  • We removed the appendix on Gaussian phase states.

---

## Round 3 · Author Response

Dear editor,

We would like to thank you and the referees for taking the time to read our manuscript carefully. We are glad to read that the referees are supportive for publication after minor corrections.

We understand that one can consider our manuscript as lengthy. Our manuscript is organized so that the Sec. 2 is a self-contained section in which we define the model and discuss the essential results summarized on Fig. 3. The following Sec. 3 and 4 are addressed to the more technically interested reader. They are devoted to quantitative discussions of the aspects requiring additional technical tools, such as the entanglement between the qubit and the modes. We recognize that this organization of our manuscript was not clearly stated in the introduction. As a result, we modified the last paragraph of the introduction to present more clearly the organization of our manuscript along those lines.

Please find below our detailed answers to the suggestions of improvement of our manuscript with a list of changes.

We thank you for your kind consideration.
With our respectful regards,
Jacquelin Luneau,
Benoît Douçot,
and David Carpentier.

---

## Round 3 · List of Changes

• We have modified the last paragraph of the introduction to present more clearly the organization of our manuscript.

  • We add a sentence in the end of Sec. 2.1.1 including Refs [54-56] mentioning the relation between the topological coupling between two bosonic modes and a qubit, and conical intersections studied in molecular physics.

  • We now precise in section 3.1 that the adiabatic states are constructed as an asymptotic series.

  • We add a paragraph in section 3.2 including Refs [55, 56, 68, 69] discussing how our results on the geometric details of the adiabatic dynamics are related to previous works on quantum simulations of spin-orbit-induced anomalous Hall effect using trapped ions.

---

## Editorial Decision

published